# Lentils and Yeast Fibers: A New Strategy to Mitigate Enterotoxigenic *Escherichia coli* (ETEC) Strain H10407 Virulence?

**DOI:** 10.3390/nu14102146

**Published:** 2022-05-21

**Authors:** Thomas Sauvaitre, Florence Van Herreweghen, Karen Delbaere, Claude Durif, Josefien Van Landuyt, Khaled Fadhlaoui, Ségolène Huille, Frédérique Chaucheyras-Durand, Lucie Etienne-Mesmin, Stéphanie Blanquet-Diot, Tom Van de Wiele

**Affiliations:** 1UMR 454 INRAE, Microbiology, Digestive Environment and Health (MEDIS), Université Clermont Auvergne, 28 Place Henri Dunant, F-63000 Clermont-Ferrand, France; thomas.sauvaitre@uca.fr (T.S.); claude.durif@uca.fr (C.D.); khaled.fadhlaoui@uca.fr (K.F.); fchaucheyrasdurand@lallemand.com (F.C.-D.); lucie.etienne-mesmin@uca.fr (L.E.-M.); 2Center for Microbial Ecology and Technology (CMET), Faculty of Bioscience Engineering, Ghent University, Coupure Links 653, B-9000 Ghent, Belgium; florence.vanherreweghen@ugent.be (F.V.H.); karen.delbaere@ugent.be (K.D.); josefien.vanlanduyt@ugent.be (J.V.L.); tom.vandewiele@ugent.be (T.V.d.W.); 3HARi&CO, 10 Rue Pierre Semard, F-69007 Lyon, France; shuille@hari-co.fr; 4Lallemand SAS, 19 Rue des Briquetiers, BP 59, CEDEX, F-31702 Blagnac, France

**Keywords:** dietary fiber, enterotoxigenic *Escherichia coli*, virulence, mucus, fecal microbiota, innate immune response

## Abstract

Dietary fibers exhibit well-known beneficial effects on human health, but their anti-infectious properties against enteric pathogens have been poorly investigated. Enterotoxigenic *Escherichia coli* (ETEC) is a major food-borne pathogen that causes acute traveler’s diarrhea. Its virulence traits mainly rely on adhesion to an epithelial surface, mucus degradation, and the secretion of two enterotoxins associated with intestinal inflammation. With the increasing burden of antibiotic resistance worldwide, there is an imperious need to develop novel alternative strategies to control ETEC infections. This study aimed to investigate, using complementary in vitro approaches, the inhibitory potential of two dietary-fiber-containing products (a lentil extract and yeast cell walls) against the human ETEC reference strain H10407. We showed that the lentil extract decreased toxin production in a dose-dependent manner, reduced pro-inflammatory interleukin-8 production, and modulated mucus-related gene induction in ETEC-infected mucus-secreting intestinal cells. We also report that the yeast product reduced ETEC adhesion to mucin and Caco-2/HT29-MTX cells. Both fiber-containing products strengthened intestinal barrier function and modulated toxin-related gene expression. In a complex human gut microbial background, both products did not elicit a significant effect on ETEC colonization. These pioneering data demonstrate the promising role of dietary fibers in controlling different stages of the ETEC infection process.

## 1. Introduction

The food- and water-borne enterotoxigenic *Escherichia coli* (ETEC) is the primary agent responsible for travelers’ diarrhea, with hundreds of millions of diarrheal episodes worldwide [1]. The site of action for ETEC is mostly localized in the distal part of the human small intestine [2,3,4]. There, a myriad of virulence factors support its infectious cycle [5,6]. The mucus-degrading proteins (YghJ and EatA) and adhesins (such as FimH and Tia) facilitate ETEC’s access to the epithelial brush border and promote ETEC attachment, respectively [7,8]. Then, ETEC’s close proximity to the intestinal epithelium favors the action of the heat-labile (LT) and/or heat-stable (ST) toxins. These enterotoxins trigger water and ion secretions in the intestinal lumen, leading to cholera diarrhea-like symptoms [9,10]. In parallel, an inflammatory response elicited by different pro-inflammatory serologic and fecal markers [11,12] and changes in microbiota composition [13,14] is also reported in infected patients, suggesting its role in ETEC physiopathology.

To date, the treatment for ETEC-associated diarrhea is the same as for any acute secretory diarrheal disease. Antibiotic therapy is often prescribed to patients, although it is not routinely recommended due to the rise in antimicrobial resistance and potential side effects on human health [15]. Potential alternative strategies are therefore being investigated, including bacteriophages [16], probiotics [17], or less well known dietary fibers [18,19].

Dietary fibers are generally defined as carbohydrate polymers with 10 or more monomeric units that are not hydrolyzed by the endogenous enzymes in the small intestine of humans, thus serving as preferential substrates for gut microbes [20]. They can be divided into subgroups according to their origins, structures, and physicochemical properties [21,22]. Most of the dietary fibers consumed by humans are of plant origin (e.g., from vegetables, legumes, or cereals), but some of them are also derived from animal products (e.g., milk), fungi, or bacteria [21]. They are subdivided into soluble and insoluble fibers: the soluble fraction is degraded faster by the microbes in the gastrointestinal tract, while the insoluble one is less accessible and constitutes a physical microbial attachment surface during intestinal transit [22,23,24]. Dietary fibers have well-known beneficial health effects in humans, such as transit regulation, slowing down glucose absorption, immune system modulation, and the support of gut microbiota diversity [25,26].

More recently, in vitro studies have also shown the antagonistic properties of fibers against various enteric bacterial pathogens, mostly through a direct bacteriostatic effect [27,28,29,30], anti-adhesion properties on intestinal cells, or a decoy for pathogen/toxin binding to mucosal polysaccharides [19,31,32,33,34]. The effects from dietary fiber can also be indirectly mediated by gut microbiota modulation, e.g., by supporting probiotic species showing anti-infectious properties [35]. Recently, a novel potential mechanism of action was also suggested. By presenting another nutrient source to the resident microbes, fibers could also lure them from mucus consumption, thereby impeding pathogen access to the underlying epithelium [36,37]. Since ETEC needs to interact with the mucus layer to fulfil its infection cycle, this strategy could be particularly relevant to investigate. However, up to now, studies specifically addressing how dietary fiber affects ETEC strains of human origin are scarce. Only milk oligosaccharides and plantain soluble fibers were proven to reduce the adhesion of ETEC strains Pb-176, CECT 685, and C410 to human Caco-2 intestinal epithelial cells [18,38,39]. Facing this lack of data, we previously conducted a short screening program to select the most promising fiber-containing products among eight candidates, i.e., a homemade lentil extract and specific yeast cell walls from *Saccharomyces cerevisiae* AQP 12,260 [19].

The aim of the present study was to investigate more deeply the anti-infectious potential of these fiber products against the human ETEC H10407 reference strain. By using complementary in vitro approaches simulating the human gastrointestinal tract, we investigated the direct and indirect effects of lentil- and yeast wall fiber-containing products on various stages of ETEC physiopathology, namely, bacterial growth, adhesion to mucus and intestinal epithelial cells, toxin production and the regulation of main virulence genes, the impact on intestinal barrier integrity, the induction of innate immunity, and human gut microbiota modulation.

## 2. Materials and Methods

### 2.1. Preparation and Characterization of Dietary-Fiber-Containing Products

The specific yeast cell wall from *Saccharomyces cerevisiae* AQP 12,260 and the raw green lentils were provided by Lallemand SAS (Blagnac, France) and HARi&CO (Lyon, France), respectively. The lentils were prepared according to their usual household consumption and were extracted by a digestion step followed by ethanol precipitation, as previously described [19]. Briefly lentils were boiled (30 min), washed in sterile distilled water, ground at maximum speed in a blender 8010S (Waring, Stamford, CT, USA) until homogeneity, and filtered through a 0.9 mm diameter pore filter. For each 200 g of raw products, 10 g of pancreatin (P1750, Merck, Darmstadt, Deutschland) was added to 200 mL of sterile distilled water and centrifuged (8000× *g*, 30 min, 4 °C). The supernatant was collected and added to the ground material with 3.2 mg of trypsin (T0303, Merck, Darmstadt, Deutschland). Digestion was performed for a total duration of 6 h (37 °C, 100 rpm). To precipitate soluble fibers, three volumes of 96% ethanol were then added to the mixture under agitation (4 °C, 100 rpm, 1 h). The solution was centrifuged (2500× *g*, 15 min, 4 °C), and the pellet was washed three times in 75% ethanol. Finally, the pellet was dried in an incubator (42 °C, overnight) and then finely ground at full speed under sterile conditions in a blender 8010S (Waring, Stamford, USA). The lentil extract was found to be sterile by plating on plate counting agar. The specific yeast cell walls were autoclaved (121 °C, 15 min) prior to use in in vitro experiments. The detailed nutritional analysis of the two fiber-containing products used in this study was performed by an external company (CAPINOV, Landerneau, France), and the results are provided in Table 1. In all experiments, the products were used at the final fiber concentration of 2 g·L^−1^ unless otherwise stated.

### 2.2. ETEC Strain and Growth Conditions

The prototypical ETEC strain H10407 serotype O78:H11:K80 (ATCC^®^ 35401, LT^+^, ST^+^, CFA/I^+^), isolated in Bangladesh from a patient with a cholera-like syndrome [40], was used in this study. Bacteria were routinely grown under agitation (37 °C, 120 rpm, overnight) in Luria–Bertani (LB) broth.

### 2.3. Growth Kinetics Assays in Broth Media

ETEC strain H10407 (initial concentration of 10^7^ CFU·mL^−1^) was allowed to grow aerobically (37 °C, 100 rpm, 5 h,) in complete LB or M9 minimal media (Sigma, St. Louis, MO, USA) with or without each fiber-containing product (2 g·L^−1^). The media were regularly sampled and plated onto LB agar for ETEC numeration (*n* = 3).

### 2.4. LT Toxin Measurement in Broth Media

The LT concentration was assayed by cultivating ETEC strain H10407 in Casamino acids-yeast extract (CAYE) medium (37 °C, 100 rpm) with or without fiber-containing products at various concentrations (ranging from 0.0625 to 8 g·L^−1^). After overnight culture, the medium was centrifuged (3000× *g*, 5 min, 4 °C), and toxin concentrations were measured in the supernatant by GM-1 ELISA assay, as previously described [19]. Pure LT toxin detection inhibition assays were also carried out as aforementioned with pure LT-Cholera B toxin sub-unit (Sigma-Aldrich, Saint-Louis, MO, USA) added in CAYE medium at a concentration of 500 ng·mL^−1^ without ETEC bacteria. The absence of a negative effect of various doses of fiber-containing products on ETEC growth was verified by plating on LB agar plates at the end of the LT experiments. Three independent biological replicates were performed for each assay.

### 2.5. Mucin Bead Adhesion Assays

Mucin–alginate beads were obtained as already described [41]. Briefly, the mixture containing 5% (*w*/*v*) porcine gastric mucin type III and 2% (*w*/*v*) sodium alginate (Sigma-Aldrich, Saint-Louis, MO, USA) was dropped using a peristaltic pump into a sterile solution of 0.2 M CaCl_2_ under agitation (100 rpm). The beads were stored at 4 °C for no more than 24 h prior to experiments. For yeast–alginate beads, mucin was replaced by the specific yeast cell wall product at the same concentration (5% *w*/*v*). Adhesion assays on beads were carried out as follows: ETEC was inoculated at a dose of 10^7^ CFU·mL^−1^ and allowed to adhere during a 1 h contact period. At the end of the experiment, beads were washed three times with ice-cold sterile physiological water and crushed with an Ultra-Turrax apparatus (IKA, Staufen, Germany). The resulting suspensions were then serially diluted and plated onto LB agar plates for ETEC numeration (“adhered” cells). In order to test adhesion inhibition by mannose residues, D-mannose (Sigma-Aldrich, Saint-Louis, MO, USA) was added at a final concentration of 10 g·L^−1^ to the medium prior to ETEC inoculation. Three independent biological replicates were performed.

### 2.6. Caco-2 and HT29-MTX Cell Culture Assays

Caco-2 and HT29-MTX cells were cultivated as already reported [19]. The Caco-2/HT29-MTX co-culture (ratio 70:30) was maintained for 18 days to reach the full differentiation stage [42]. Cells were pre-treated or not with fiber-containing products (2 g·L^−1^) for a 3 h period. The cells were then infected with ETEC strain H10407 at a multiplicity of infection (MOI) of 100 for 3 additional hours (37 °C, 5% CO_2_) in antibiotic/antimycotic-free medium. At the end of experiment, to monitor ETEC “planktonic” bacteria cells, culture medium was collected and centrifuged (3000× *g*, 5 min, 4 °C). The resulting pellet was kept in RNA later (Invitrogen, Waltham, MA, USA) at −20 °C for downstream RNA extraction and RT-qPCR analysis of ETEC virulence genes. To monitor ETEC “adhered” bacteria, cell layers were washed three times with ice-cold PBS (ThermoFisher, Waltham, MA, USA). In a first set of experiments, Caco-2/HT29-MTX cells were lysed with 1% Triton X-100 (Sigma-Aldrich, Saint-Louis, MO, USA). Cell lysates were plated onto LB agar to determine the number of ETEC bacteria adhered to the cells or further centrifuged (3000× *g*, 5 min, 4 °C). The resulting supernatant was used to measure the intracellular pro-inflammatory IL-8 levels, while pellet cells were stored in RNA later (Invitrogen, Waltham, MA, USA) at −20 °C for further prokaryote RNA extraction. In a second set of experiments, RNAs from adhered bacteria were extracted for eukaryotic gene expression analysis (ETEC virulence genes). Control experiments were also performed with non-infected Caco-2/HT29-MTX cells and in the DMEM medium devoid of intestinal cells for virulence gene expression analysis. The impact of both ETEC strain H10407 and fiber-containing products on intestinal cell viability was controlled during a 3 h time course using a Trypan blue exclusion assay. For each set of experiments, at least three independent biological replicates were performed.

### 2.7. Measurement of Caco-2/HT29-MTX Permeability on Transwells

For permeability experiments, Caco-2/HT29-MTX cells were rinsed with PBS and incubated with an apical concentration of caffeine (1 g·L^−1^) or atenolol (50 mg·L^−1^) in fresh DMEM medium with or without dietary-fiber-containing products (2 g·L^−1^). The medium was collected after 2 h of incubation at both the apical and basolateral sides of the transwells. The caffeine and atenolol concentrations were determined by HPLC (Elite LaChrom, Merck HITACHI, USA) using an Onyx™ Monolithic C18 LC column of 100 × 4.6 mm at 20 °C (Phenomenex, Torrance, CA, USA) and an Interchimm C18 column of 250 × 4.6 mm at 40 °C (Interchim, Montluçon, France), respectively. The mobile phase was composed of acetonitrile/pH 6.5 PBS (10:90, *v*/*v*) and acetonitrile/water (20:80, *v*/*v*) with 10 mM ammonium acetate for caffeine and atenolol, respectively. Data were obtained and analyzed by the EZChrom Elite software at 235 and 275 nm for caffeine and atenolol, respectively. The caffeine and atenolol concentrations were calculated from standard curves established from known serial dilutions of each compound. The molecular absorption was defined as the percentage of basal molecules/total molecules introduced. The transepithelial resistance (TEER) was measured regularly during the time course of the experiment (total duration = 3 h) with a volt/ohmmeter (World Precision Instruments, Hessen, Germany). Three independent biological replicates were performed.

### 2.8. RNA Extractions

Eukaryotic RNAs were extracted with the RNeasy Plus Mini Kit (Qiagen, Germany). Total bacterial RNAs were extracted using the TRIzol^®^ method (Invitrogen, Waltham, MA, USA), as already described [43], with an additional purification step with a MinElute Cleanup Kit (Qiagen, Hilden, Deutschland). The nucleic acid purity was checked and RNA was quantified using the NanoDrop ND-1000 (Thermo Fisher Scientific, Waltham, MA, USA). To remove any contamination by genomic DNA, a DNAse treatment was performed [43].

### 2.9. Quantitative Reverse Transcription (RT-qPCR) Analysis of ETEC Virulence Genes

cDNA amplification was achieved using a CFX96 apparatus (Bio-Rad, Hercules, CA, USA), and q-PCR was performed using the primers listed in Table 2. qPCR data were analyzed using the comparative E^−ΔΔCt^ method and were normalized with the reference genes *tufA* and *ihfB*. The amplification efficiency of each primer pair was controlled from the slope of the standard curves (E = 10^(−1/slope)^-1), based on a serial dilution of a pool of three ETEC cDNA samples. Differences in the relative expression levels of each virulence gene were calculated as follows: ΔΔCt = (Ct_target gene_ − Ct_reference gene_)_in the tested condition_ – (Ct_target_ gene − Ct_reference gene_)_in the reference condition_, and data were derived from E^−ΔΔCt^.

### 2.10. Quantitative Reverse Transcription (RT-qPCR) Analysis of Selected Intestinal Cell Genes

Reverse transcriptions were first performed with the High-Capacity cDNA Reverse Transcription Kit (Thermo Fisher Scientific, Waltham, MA, USA). Then, quantitative PCR was carried out using PowerUp™ SYBR™ Green Master Mix (Thermo Fisher Scientific, Waltham, MA, USA) and a TaqMan 7900 Fast instrument (Thermo Fisher Scientific, Waltham, MA, USA) with the primers listed in Table 2. The expression of host genes related to mucin synthesis, tight junction proteins, and inflammation were investigated. The data were analyzed with SDS 2.3 software (Thermo Fisher Scientific, Waltham, MA, USA) using the comparative 2^−ΔΔCt^ method and were normalized with the reference genes *GAPDH, HPRT,* and *PPIA*. The amplification efficiency of each primer pair was controlled from the slope of the standard curves (E = 10^(−1/slope)^-1) based on a serial dilution of a pool of six RNA samples from the experiments.

### 2.11. Measurement of Interleukin-8 by ELISA

Pro-inflammatory IL-8 cytokine concentrations were determined in cell lysates from the Caco-2/HT29-MTX co-culture experiments according to the manufacturer’s instructions (DuoSet ELISA, human CXCL8/IL-8, RnD Systems, Minneapolis, MN, USA). The results were expressed as fold changes compared to control experiments performed without ETEC (non-infected) or fiber-containing products (non-treated).

### 2.12. Batch Experiments

Batch experiments were carried out for 24 h in 60 mL penicillin bottles containing 20 mL of nutrient medium and 60 mucin–alginate beads. Each liter of medium was composed of: 1 g of potato starch, 1 g of yeast extract, 1 g of proteose peptone, and 1 g of type III pig gastric mucin (all from Sigma Aldrich, St. Louis, MO, US) suspended into 0.1 M phosphate buffer (pH 6.8) and autoclaved before use. The lentil extract and yeast cell wall products were added at a final fiber concentration of 2 g·L^−1^_._ In the control condition with no dietary-fiber-containing product (non-treated), the composition of the nutritive medium was compensated by the addition of 0.5 g of guar gum, 1 g of pectin, and 0.5 g of xylan (same total fiber concentration).

To examine the inter-individual variability of ETEC interactions with dietary-fiber-containing products and human gut microbiota, the experiments were replicated with fecal samples from six healthy individuals. These donors were three males (donors 1, 2, and 3) and three females (donors 4, 5, and 6), ranging in age from 20 to 30 years, without a history of antibiotic use six months prior to the study. Consent for fecal collection was obtained under registration number BE670201836318 (Gent University). The fecal collection and fecal slurry preparation were performed as previously described [61]. An inoculation at a 1:5 dilution of the 20% (*w*/*v*) fecal slurry resulted in a final concentration of 4% (*w*/*v*) fecal inoculum in the penicillin bottles. To reproduce stresses that the pathogens endure during transit in the stomach and small intestine in humans, ETEC strain H10407 was pre-digested using a simple static gastrointestinal procedure, as described in Table 3 and already described [43]. ETEC was inoculated at the final concentration of 10^8^ CFU·mL^−1^. The penicillin bottles were flushed with N_2_/CO_2_ (80%/20%) during 20 cycles to obtain anaerobic conditions. The cycle was stopped at overpressure, and before the start of the experiment, the bottles were set at atmospheric pressure. The penicillin bottles were incubated (37 °C, 120 rpm) on a KS 4000i orbital shaker (IKA, Staufen, Germany) and aliquots were taken immediately after the start of the incubation (T0) and at 24 h of fermentation (T24h) from the liquid and atmospheric phases. Mucin–alginate beads were collected 24 h post-inoculation and were washed twice in ice-cold physiological buffer before storage. All aliquots were immediately stored at −20 °C, except samples for flow cytometry that were fixed before storage.

### 2.13. Gut Microbiota Metabolite Analysis

Short chain fatty acid (SCFA) production was measured using capillary gas chromatography coupled to a flame ionization detector after diethyl ether extraction as previously described [61,62]. The gas phase composition was analyzed with a Compact gas chromatograph (Global Analyser Solutions, Breda, The Netherlands) equipped with a Molsieve 5A pre-column and Porabond column (CH_4_, O_2_, H_2,_ and N_2_) or an Rt-Q-bond pre-column and column (CO_2_). The concentrations of gases were determined with a thermal conductivity detector. The total pressure in the penicillin bottles was analyzed using a tensiometer (Greisinger, Regenstauf, Germany).

### 2.14. DNA Extraction

DNA extraction and quality controls were performed from samples collected at T0 and T24h during batch experiments as previously described [61,63]. The DNA quality and quantity were verified by electrophoresis on a 1.5% (*w*/*v*) agarose gel and an analysis on a spectrophotometer DENOVIX ds-11 (Denovix, Wilmington, DE, USA).

### 2.15. ETEC Quantification by qPCR

qPCR was performed using a StepOnePlus real-time PCR system (Applied Biosystems, Waltham, MA, USA). The reactions were conducted in a total volume of 20 μL, consisting of 10 μL of 2× iTaq universal SYBR Green supermix (Bio-Rad Laboratories, Hercules, CA, USA), 2 μL of DNA template, 0.8 μL (10 µM stock) of each primer, and 6.4 μL of nuclease-free water. The primers used for ETEC quantification are listed in Table 2. The data were analyzed using the comparative E^−ΔΔCt^ method. The amplification efficiency of the primer pairs was determined by the generation of a standard curve based on the serial dilution of five ETEC-infected samples. Differences in the number of copies of the *eltB* gene were calculated as follows: ΔΔCt = (Ct_target gene_ – Ct_reference gene_)_sample of interest_ – (Ct_target gene_ – Ct_reference gene_)_reference sample_, and data were derived from E^-ΔΔCt^. All qPCR analyses were conducted in triplicate.

### 2.16. ETEC Quantification by RNA Fluorescent In Situ Hybridization

Flow cytometry samples were fixed and prepared for RNA fluorescent in situ hybridization, as already described [64]. Cells were hybridized in 100 µL of hybridization buffer for 3 h at 46 °C. The hybridization buffer consisted of 900 mmol·L^−1^ NaCl, 20 mmol·L^−1^ Tris–HCl (pH 7.2), 0.01% sodium dodecyl sulfate, 20% deionized formamide, and 5 mM EDTA. The buffer also contained the two *E. coli*-targeting probes at a final concentration of 2 ng·µL^−1^ and a combination of probes targeting eubacteria at a final concentration of 1 ng·µL^−1^ each (Table 2). After hybridization, the samples were washed with wash buffer (900 mmol.L^−1^ NaCl, 20 mmol.L^−1^ Tris–HCl pH 7.2, 0.01% sodium dodecyl sulfate) for 15 min at 48 °C. After washing, the cells were resuspended in 50 µL of PBS. The samples were diluted and stained with SYBR^®^ Green I (100× concentrate in 0.22 μm filtered dimethyl sulfoxide, Invitrogen) and incubated for 20 min at 37 °C. The samples were analyzed immediately after incubation with an Attune NxT BRXX flow cytometer (Thermo Fisher Scientific, Waltham, MA, USA). The flow cytometer was operated with Attune™ Focusing Fluid as the sheath fluid. The threshold was set on the primary emission channel of blue lasers (488 nm). The Attune Cytometric Software was used to draw the gates, and the percentage of active *E. coli* in the total bacteria population was expressed as the number of cells showing the *E. coli* probe fluorescence out of the number of cells fluorescently labeled with the Eubacteria probes and SYBR green fluorescence.

### 2.17. 16S Metabarcoding Analysis of Gut Microbial Communities

Next-generation 16S rRNA gene amplicon sequencing of the V3-V4 region was performed by LGC Genomics (Berlin, Germany) on an Illumina MiSeq platform (Illumina, San Diego, CA, USA), as previously described [61], except that the luminal and mucosal samples had undergone 30 and 33 amplification cycles, respectively.

All data analysis was performed in R (4.1.2). The DADA2 R package was used to process the amplicon sequence data according to the pipeline tutorial [65]. In a first quality control step, the primer sequences were removed, and reads were truncated at a quality score cut-off (truncQ = 2). Besides trimming, additional filtering was performed to eliminate reads containing any ambiguous base calls or reads with a high number of expected errors (maxEE = 2.2). After dereplication, the unique reads were further denoised using the DADA error estimation algorithm and the selfConsist sample inference algorithm (with option pooling = TRUE). The obtained error rates were further inspected, and after approval, the denoised reads were merged. Subsequently, the ASV table obtained after chimera removal was used for taxonomy assignment using the Naive Bayesian classifier, and the DADA2-formatted Silva v138 ASVs mapping back to anything other than ‘Bacteria’ as well as singletons were excluded and considered to be technical noise [66].

### 2.18. Statistical Analysis

All statistical analyses, except the one conducted on the microbiota diversity composition results, were performed using GraphPad Prism v8.0.1. The statistical data analysis on microbiota diversity was performed using R, version 4.1.2 (R Core Team, 2016), using the statistical packages Phyloseq (v1.38) [67] for ASV data handling, vegan v2.5.7 [68], betapart v 1.5.4 for diversity analysis of ASV’s [69] and deseq2 v1.34 [70] for the significant higher/lower abundance of ASVs. The evolution of the microbial community α-diversity between conditions was followed by computing the richness (observed ASV) and evenness indexes (Shannon, Simpson, inverse Simpson, and Fisher) using vegan. To highlight the differences in microbial community composition between conditions, ordination and clustering techniques were applied and visualized with ggplot2 (v3.3.5) [71]. Non-metric multidimensional scaling (NMDS) was based on the relative-abundance-based Bray–Curtis dissimilarity matrix [72]. The influence of ETEC infection and the type of beads used was determined by applying a distance-based redundancy analysis (db-RDA) using the abundance-based Bray–Curtis distance as a response variable [71,73]. db-RDA was performed both including and excluding ASV1 (attributed to *Escherichia*/*Shigella*) from the ASV table. The significance of group separation between conditions was also assessed with a permutational multivariate analysis of variance (permANOVA) using distance matrixes [71]. Prior to this formal hypothesis testing, the assumption of similar multivariate dispersions was evaluated. In order to find statistically significant differences in ASV abundance between the infected and non-infected conditions, a Wald test (corrected for multiple testing using the Benjamini and Hochberg method) was applied using the DESeq2 package. The metabolic response (measured SCFA and pH) was modeled in the function of the beads and infection conditions in a db-RDA analysis.

## 3. Results

### 3.1. Fiber-Containing Products Do Not Impede ETEC Growth in Complete Culture Medium

When ETEC strain H10407 was grown in an LB-rich medium (Figure 1A), no statistical difference was observed between the conditions supplemented with the lentil extract (‘lentils’) and the specific yeast cell walls (‘yeast’) compared to the negative control (‘non-treated’). Therefore, neither of the two fiber-containing products were able to impede ETEC growth in a nutrient-rich culture medium. In M9 minimal medium (Figure 1B), both products were able to sustain ETEC growth compared to the non-treated condition, leading to an almost 2-log difference with the control condition after 5 h of incubation. This overgrowth became statistically different at 240 and 300 min according to Dunnett’s multiple comparisons test (*p* < 0.05).

### 3.2. The Lentil Extract Decreases LT Toxin Concentrations in a Dose-Dependent Manner

Irrespective of the dose tested, specific yeast walls had no effect on LT toxin concentrations (Figure 2A). In contrast, the lentil extract significantly decreased LT toxin concentrations in a clear dose-dependent manner (Figure 2B). This inhibitory effect was significant, starting at the dose of 0.065 g·L^−1^ (1.64-fold decrease, *p <* 0.05). LT toxin was no longer detected when the lentil concentration exceeded 1 g·L^−1^. To further investigate the possible mechanism of inhibition, we incubated the pure B sub-unit of the LT toxin at 500 ng.mL^−1^ with various doses of the lentil extract in the absence of ETEC (Figure 2C). The lentil extract tended to inhibit LT toxin detection by the GM1-ELISA assay in a dose-dependent manner. At the highest fiber dose tested (8.0 g·L^−1^), the LT concentrations were 36-fold lower (6.0 ± 9.1 ng·mL^−1^) compared to the lowest dose (0.0625 g·L^−1^, 214.8 ± 158.9 ng.mL^−1^, *p* = 0.08). Finally, we verified that the lentil extract had no effect on ETEC growth in the CAYE medium during the LT assays (Figure 2D).

### 3.3. Yeast Cell Walls Inhibit ETEC Adhesion to Mucin and Mucus-Secreting Intestinal Cells

First, the absence of a deleterious effect of both ETEC strain H10407 and fiber-containing products on intestinal cell viability was confirmed (Appendix A). The lentil extract and yeast cell walls were able to significantly reduce ETEC adhesion to mucin–alginate beads by about 6- and 3-fold, respectively (Figure 3A, *p <* 0.05). Compared to the control condition, the yeast cell walls reduced the number of adhered ETEC bacteria to Caco-2/HT29-MTX cells by nearly one log compared to the non-treated condition (*p <* 0.001, Figure 3B). Additional experiments were performed with yeast–alginate beads to challenge ETEC’s affinity for yeast cell walls (Figure 3C). ETEC adhesion on yeast–alginate beads was significantly increased compared to mucin–alginate beads (nearly a one-log increase, *p <* 0.01). The addition of mannose at 10 g·L^−1^ in the medium did not affect ETEC adhesion on yeast–alginate beads (non-significant 33% inhibition, *p >* 0.05), while it had a significant impact on the number of adherent bacteria on mucin–alginate beads (64% inhibition, *p <* 0.01).

### 3.4. Both Fiber-Containing Products Modulate ETEC Toxin-Related Virulence Gene Expression

The impact of the fiber-containing products on ETEC strain H10407 virulence genes was analyzed using two different experimental set-ups: with Caco-2/HT29-MTX cells (Figure 4A) or in the DMEM medium devoid of intestinal cells (Figure 4B). Overall, the lentil extract and specific yeast cell walls both had a strong effect on the virulence gene expression of planktonic ETEC bacteria (i.e non-adhered) whether in the presence or absence of intestinal cells. Interestingly, the lentil extract upregulated the expression of *fimH* adhesin (5.3- to 8.6-fold) and *YghJ* mucinase (2.3- to 10.3-fold) genes (Figure 4A,B) while also downregulating the expression of the two toxin genes *eltB* and *estp* as well as *tolC,* which participates in ST toxin secretion and the *rpoS* gene involved in environmental stress responses. The presence of intestinal cells did not impact the modulatory effect of lentils towards ETEC gene expression. The yeast cell walls increased the expression of the two adhesins, *fimH* and *tia*, as well as the genes involved in LT toxin production and secretion, *eltB* and *leoA*, from 1.32- to 4.47-fold, depending on the genes (Figure 4A, B). In the non-treated conditions, cell adhesion increased virulence gene expression, as reported by the *fimH, eltB,* and *estP* respective 5.5-, 2.3-, and 3.0-fold increases (*p* < 0.05, Figure 4A). Compared to planktonic bacteria, the modulation of adhered bacteria virulence by dietary-fiber-containing products was more subtle (Figure 4A). The two compounds reduced *eltB* and *estP* toxin gene induction to a maximum of 1.7-fold compared to the non-treated control (Figure 4A). In particular, yeast walls significantly reduced *estP* gene induction in adhered bacteria by 90% (*p* < 0.05). In contrast, none of the fiber products succeeded in reducing the 5-fold *fimH* induction by cell adhesion (Figure 4A), with a slight promoting effect for yeast cell walls (1.28-fold increase, *p* < 0.05). Lastly, both the lentil extract and yeast walls tended to reduce the environmental stresses encountered by adhered ETEC, as reported by the respective 60% and 70% decreases in *rpos* expression (Figure 4A).

### 3.5. The lentil Extract Limits ETEC-Induced Inflammation

Host innate immune response-related genes (cytokines) were selected and analyzed during the Caco-2/HT29-MTX experiments. ETEC infection of intestinal cells triggered the expression of all cytokine genes, as reported by the respective 65-, 5-, 63-, 244-, and 2-fold increases in *TNF-α*, *IL-1β*, *IL-6*, *IL-8,* and *IL-10* expression (*p <* 0.05, Figure 5). The lentil extract tended to reduce the induction of all of these genes, with significance reached for *IL-1β*, *IL-6,* and *IL-10* (*p <* 0.05), with decreases of 52, 52, and 41%, respectively (Figure 5A, 5C, and 5D). The results were more mitigated with the specific yeast cell walls, which only reduced *IL-10* expression (*p <* 0.01) (Figure 5A). We further analyzed the IL-8 concentration to assess the impact of fiber-containing products on cytokine induction at the protein level. As expected, ETEC inoculation induced a significant (*p <* 0.001) 1.6-fold increase in intracellular IL-8 production (Figure 5F). Both the yeast walls and lentil extract were able to significantly decrease IL-8 intracellular production under non-infected conditions (*p <* 0.05). In the infected condition, the protective effect was mostly preserved for lentils (*p <* 0.001), with relative IL-8 levels comparable to the control condition without any fiber or bacteria (0.85 ± 0.07 versus 1.00 ± 0.12), while the results obtained with yeast walls almost reached significance (*p =* 0.06).

### 3.6. The Lentil Extract Modulates ETEC Induction of Mucus-Related Gene Expression

Furthermore, mucus-related gene expression was assayed as a witness of the innate effector response. Inoculation with ETEC strain H10407 tended to induce all selected genes except *TTF3* (Figure 6). This induction was significant (*p <* 0.05) for *MUC17* (3-fold) and *KLF4* (2-fold) only. The lentil extract tended to mitigate the ETEC induction of *MUC1*, *MUC2*, *MUC5AC*, *MUC5B,* and *KLF4,* with significance reached for *MUC1* and *KLF4* (*p <* 0.05, Figure 6). *MUC1* and *KLF4* expression were induced by 1.5- and 2.3-fold under the infected condition and returned at 0.9- and 1.2-fold of their basal expression levels with the lentil extract, respectively (Figure 6A,H). Contrarily, the lentil extract favored the basal expression of *MUC17* (2.4-fold induction, *p <* 0.05), and this effect was conserved after ETEC inoculation (1.3-fold compared to the non-treated control, *p <* 0.05, Figure 6F).

### 3.7. Yeast Cell Walls Strengthen Intestinal Barrier Function

As human ETEC strains and their virulence factors can potentially impact the epithelial barrier, the expression of tight-junction-related genes was also followed during the cellular experiments. Among the four genes that were studied (Figure 7), only *CLDN1* was significantly induced by ETEC infection (1.6-fold induction, *p <* 0.05). Interestingly, this induction was reduced by the yeast cell walls to almost return to the basal level (*p <* 0.05, Figure 7A). *TJP1* expression was also triggered by the lentil extract but only when ETEC strain H10407 was inoculated (3.4-fold induction, *p* < 0.05, Figure 7C). Considering these mitigated results, we decided to assess the effect of fiber-containing products on epithelial barrier permeability. When applied to the apical side of Caco-2/HT29-MTX transwells, after 2 h of contact, none of the tested products increased the absorption of caffeine (Figure 7E) or atenolol (Figure 7F), which were used as markers for transcellular and paracellular permeability [74,75], respectively. Yeast cell walls even significantly decreased caffeine absorption from 21.2 to 17.0% (*p* < 0.05, Figure 7E), and both products strongly reduced (*p* < 0.05) atenolol absorption, with 3.0- and 5.8-fold reductions for the lentil extract and yeast cell walls, respectively (Figure 7F). Accordingly, fiber-containing products led to a rise in TEER over time, with significant 1.3- and 1.4-fold increases for the lentil extract compared to the non-treated condition at 120 and 180 min, respectively (*p* < 0.05, Figure 7G).

### 3.8. Yeast Cell Walls Mostly Impact Mucus-Associated Microbiota during ETEC Infection

To investigate the impact of dietary-fiber-containing products on ETEC interactions with luminal and mucosal gut microbiota, batch experiments inoculated with human feces were performed in flasks containing mucin–alginate beads. As expected, at the start of the experiment, the *Escherichia*/*Shigella* population became predominant in the luminal phase of infected bottles and represented 34% of the read count detected by *16S rRNA* gene sequencing (Figure 8C) and 15% of active bacteria by RNA fluorescent in situ hybridization (Figure 8E). The proportion of ETEC or *Escherichia*/*Shigella* in the luminal phase remained stable during the experimental time course, regardless of the detection technique used (Figure 8A,C,E). Dietary-fiber-containing products had no significant effect on *Escherichia*/*Shigella* or ETEC proportions in the luminal phase (Figure 8A,C,E), but a decreasing trend in ETEC levels (1.7-fold lower) with yeast cell walls was observed (Figure 8A). Concerning the mucosal compartment, in infected conditions, the number of adherent ETEC, as reported by qPCR, tended to be, respectively, 1.2- and 1.7-fold lower with the lentil extract and yeast cell walls compared to the non-treated control, but again, no significance was reached (Figure 8B). The *16S rRNA* gene sequencing showed a non-significant 33% decrease in adhered *Escherichia*/*Shigella* ASV under the yeast cell wall condition compared to the non-treated one (Figure 8D).

### 3.9. Fiber Products Have No Significant Effect on ETEC Colonization in a Complex Microbial Background

To further explore the effects of dietary-fiber-containing products on gut microbiota composition, we performed *16S rRNA* gene sequencing and bacterial community analysis. Regarding α-diversity, ETEC infection was associated with a significant decrease in α-diversity evenness in the luminal phase, but supplementation with fiber-containing products had no effect (Figure 9B,C). Both infection by ETEC and supplementation with fiber products had no influence on species richness in the luminal phase (Figure 9A) and on both species’ richness and evenness in the mucosal phase (Figure 9D–F).

Concerning β-diversity, an NMDS analysis showed that the stool donor was the predominant explanatory variable for dissimilarities in gut microbiota composition in both the luminal and mucosal compartments (Figure 10A). A PermANOVA analysis performed on the samples at T24h and excluding ASV1 (attributed to *Escherichia*/*Shigella*) confirmed that donor origin accounted for 10.0% of the dissimilarities (*p* < 0.001, 999 permutations). ETEC infection was also a significant source of variations and accounted for 6.0% of the dissimilarities (*p* < 0.001, 999 permutations), but dietary-fiber-containing products was not (*p* = 0.51). To go further, a db-RDA analysis was performed on samples at 24 h using “yeast”, “lentil”, and “infection” as explanatory variables. The db-RDA was able to cluster infected samples from non-infected ones more efficiently in the mucosal phase (Figure 10B). If none of the tested products were able to modify the impact of infection on the gut microbiota structure, yeast samples clustered away from the rest in both the luminal and mucosal compartments, suggesting that the yeast cell wall product was responsible for some variations in the microbiota community structure, although only modestly (Figure 10B). In the luminal phase, ETEC infection induced a global increase in *Escherichia*/*Shigella* (Figure 8A) to the detriment of other groups, such as *Bacteroides* (Figure 10C,D and Appendix A). At the genus and family levels, no clear difference in phylogenetic groups’ relative abundances was observed between the control and treated conditions at 24 h in the luminal phase, apart from a light but consistent increase in *Tannerellaceae*/*Parabacteroides* by yeast cell walls regardless of the infection status (Figure 10C,D, Appendix A). Compared to the luminal microbiota, the mucosal non-infected microbiota was depleted of *Faecalibacterium* and enriched in *Clostridium*, *Roseburia*, *Bifidobacterium,* and *Lactobacillus*, even if *Lactobacillus* colonization appeared to be donor-dependent (Figure 10C,D and Appendix A). In the non-treated condition, ETEC infection tended to be constantly detrimental to the *Clostridium* and *Bifidobacterium* species representation on mucin beads, and the dietary-fiber-containing products tended to limit the *Clostridium* disappearance (Figure 10C,D and Appendix A). In the luminal compartment, yeast cell walls seemed to reduce *Faecalibacterium* and *Ruminococcaceae* abundance and to favor *Tannerellaceae*/*Parabacteroides,* while in the mucosal compartment, they appeared to favor *Tannerellaceae*/*Parabacteroides* and commensal *Escherichia*/*Shigella* colonization. No clear trend was identified for the lentil extract (Figure 10C,D, Appendix A).

### 3.10. Fiber-Containing Products Slightly Affect Gut Microbial Activities during ETEC Infection

In a last step, the effect of dietary-fiber-containing products on gut microbial activity during ETEC infection was assessed by following various indicators, such as SCFA, gas production, pH acidification, and gas pressure. We also investigated mucin–alginate bead degradation as a measure of the mucosal microbiota degrading capability. ETEC inoculation significantly impacted butyric acid production (*p <* 0.05, two-way ANOVA), with 1.3-, 1.4- and 1.2-fold increases in the non-treated, lentils, and yeast conditions, even if no individual significances were reached (Figure 11A). When added, the lentil extract and yeast cell walls increased propionic acid production by 10–20% and 30–40%, respectively, with only the yeast condition reaching significance (*p* < 0.05, Figure 11A). Regarding pH acidification, at 24h of fermentation, the pH tended to be increased by around 0.1 when ETEC was inoculated (*p* = 0.07, two-way ANOVA), with no significant effect from fibers (Figure 11B). ETEC inoculation also tended to be associated with an increased pressure in the bottles at the end of the experiment (*p* = 0.08, two-way ANOVA, Figure 11C), with again, no significant impact of fibers. Gas analysis showed that CO_2_ levels were significantly impacted by both ETEC and fiber-containing products compared to the non-treated and non-infected control conditions (*p <* 0.05, Figure 11D). However, the addition of fiber products exhibited no significant impact on gas composition under the infected condition. Lastly, dietary-fiber-containing products led to a decrease in mucin bead weight at 24 h and reached significance for the yeast cell walls in the infected condition (*p <* 0.01, Figure 11E). Yeast supplementation was indeed associated with an increase in bead degradation by 22 and 23% in the non-infected and infected conditions, respectively. In accordance with our observations, the microbial community structure of the infected samples correlated with pH and butyric acid production, and dietary-fiber-containing products had no effect (Figure 12).

## 4. Discussion

To date, only a few studies have investigated the potential anti-infectious properties of dietary fibers against the ETEC strains responsible for traveler’s diarrhea in humans [18,38,39,76,77]. Using a large panel of complementary in vitro models, we showed that two fiber-containing products from legumes and microbes, namely, a lentil extract and a specific yeast cell wall from *Saccharomyces cerevisiae,* selected previously [19], were able to exert antagonistic effects towards the ETEC reference strain H10407 at various stages of the pathological process. These products from different origins contain various types of soluble and insoluble fibers, mainly resistant starch, cellulose, and hemi-cellulose for lentils [78] and mannans and β-glucans for yeast cell walls [79]. This variation could explain their differences in terms of the anti-infectious properties found in the present study. The two fiber products were tested at the in vivo relevant concentration of 2 grams per liter of final fiber content. This value was calculated based on the 10 to 30 grams of fibers consumed per day in industrialized countries [80,81] and the approximately 10 liters of fluid passing through the GI tract daily [82]. Of note, as the tested products were not pure, we cannot exclude that components other than fibers could exert anti-infectious properties against ETEC [83].

A first target in our study was to investigate if fiber products could affect pathogen growth in classical broth media. None of the tested compounds were able to impact the growth of ETEC strain H10407. This is not unexpected since, to our knowledge, only the human-engineered fiber chitosan has been reported to exert a bacteriostatic effect in vitro on diverse bacterial pathogens, such as enterohemorrhagic *Escherichia coli* (EHEC) [27]. We also showed that the lentil extract and yeast cell walls were able to sustain ETEC growth in M9 minimal medium, most likely due to the presence of non-fiber components, as *E. coli* strains are not able to degrade complex polysaccharides on their own [84,85]. We argue that this positive effect on pathogen growth may not be an issue in the context of the complex nutritional and microbial background of the distal small intestine, the main site of ETEC colonization [3,4,86,87,88,89]. In the human gut, fibers are degraded into smaller carbohydrates by the endogenous gut microbiota, providing substrates for pathogens, such as ETEC, which generally behave as secondary degraders [90]. By performing fecal batch experiments including microbiota from human origin, we confirmed that dietary-fiber-containing products had no significant effect on ETEC colonization in a complex microbial environment, with only a slight tendency of yeast cell walls to reduce pathogen levels in both the luminal and mucosal compartments.

As toxin production is a key feature in ETEC physiopathology, our next step was to study the impact of fiber products on LT toxin. To our knowledge, only one study has previously reported an indirect effect of dietary fibers on ETEC toxins. SCFAs, major end-products of dietary fiber metabolism by gut microbiota, have been shown to significantly reduce or even abolish LT toxin production at a concentration of 2 g·L^−1^ in CAYE culture medium [91]. Here, we showed that the LT toxin concentration was significantly reduced in culture medium by the lentil extract in a dose-dependent manner. This effect seems to be partly due to the toxin binding to some lentil components acting as a decoy, as previously reported by other groups with GM1 ELISA assays used with other carbohydrates [92]. Despite the involvement of several virulence genes in the ETEC infectious process (including those encoding for toxin production), data investigating the direct impact of dietary fibers on ETEC virulence gene expression are clearly missing in the literature. In this study, we investigated a panel of ETEC virulence genes in cellular assays. We demonstrated that such compounds could be used to modulate the induction of ETEC virulence gene expression by cellular proximity. Such induction was already reported by a previous study for ETEC strain H10407, but on non-mucus secreting Caco-2 cells [93]. Here, we showed that, at the transcriptional level, the *eltB* gene was consistently inhibited by the lentil extract. Dietary fiber supplementation is known to modulate the expression of genes involved in fiber degradation [85,94]. Only a few studies investigated the modulation of virulence genes. As an example, chitosan significantly modified *Campylobacter jejuni* genes involved in motility, quorum sensing, stress response, and adhesion [95]. Here, our study indicates that a toxin concentration decrease could be mediated by a direct inhibitory effect of the lentil extract on the LT toxin encoding gene expression.

Getting access to the epithelium is a crucial step for most intestinal pathogens to fulfill their infection cycle [96]. To this sole purpose, ETEC strain H10407 possesses two mucus-degrading enzymes [7,97] and numerous adhesins allowing mucosal adhesion [5,6]. To date, only milk oligosaccharides [38,39] and soluble plantain fibers at a dose of 5 g·L^−1^ [18] have shown the ability to reduce the adhesion of human ETEC strains (others than H10407) to a Caco-2 cell line. Here, we used a co-culture of enterocytes and mucus-secreting cells to more accurately mimic the physiological situation in the human intestine [42,98,99]. We first observed the inhibition of ETEC adhesion by both fiber-containing products on mucin beads. This anti-adhesive property cannot be explained by the sedimentation effect observed with insoluble fiber particles, as beads were always maintained under agitation. Only the yeast cell walls were able to reduce ETEC adherence in the more complex Caco-2/HT29-MTX model. Microorganism-derived polysaccharides have already shown adhesion inhibition properties against enteric pathogens [33,100,101,102], but this is the first time that yeast cell walls reduced mucosal adhesion of an ETEC strain of human origin. By using yeast–alginate beads, we showed that ETEC strain H10407 presented a greater adhesion specificity for the yeast cell walls than for mucin, supporting a potential decoy effect of the product during pathogen adhesion. However, this observed decoy effect did not seem to involve mannose residues, as previously shown when living probiotic yeasts were used [17].

ETEC, as well as its virulence factors, is well known to be linked to innate immune activation and the induction of inflammation in epithelial cell lines, animals, and humans [11,103,104,105,106,107,108], which could be positively associated with infection severity [12,109]. Here, as expected, we observed a general induction of cytokine-related genes upon ETEC H10407 exposure in cellular experiments [77]. Interestingly, the lentil extract showed a significant inhibitory effect on those genes, while the influence of yeast cell walls was more subtle. The most striking effect was observed on the pro-inflammatory *IL-8* for which inhibition by fiber products was observed not only at the gene level but also at the protein level. The underlying mechanisms of dietary fiber modulation of the innate immune response are not clear. A study from He and colleagues, performed on a human ETEC strain, showed that human milk oligosaccharide 2’-fucosyllactose could modulate CD14 expression in infected enterocytes, thus attenuating LPS-induced inflammation [77]. Here, our results showed that the products exerted a basal anti-inflammatory effect (as shown with IL-8 production) but also led to an inhibition of the innate immune response activation, regardless of the inflammatory status (as shown with IL-10 expression), which could be the result of decreased interactions with innate immune receptors. The activation of innate immune receptors is known to ultimately stimulate mucus secretion [110,111,112]. Accordingly, we found in this study that mucus-related genes tended to be activated following ETEC infection and that this activation was limited by both fiber products, with a more significant effect of the lentil extract. Of note, as mucus secretion is involved in pathogen clearance from the mucosal epithelium [111], an inhibition of mucus-related genes by the lentil extract may be considered to be unfavorable in the fight against the ETEC pathogen.

The regulation of tight junctions in intestinal epithelial cells is one of the main means for the host to control epithelial permeability [113]. ETEC ST toxin variants have been largely described as modulators of paracellular permeability and more specifically of tight junctions [114,115,116]. In contrast, few studies have investigated the effect of whole ETEC bacteria on cell permeability, most of them being performed with pig and not human strains [117,118,119]. Kreisberg and colleagues reported that some human ETEC strains, including H10407, elicited a reduction in trans-epithelial electrical resistance (TEER) in T-84 epithelial cell monolayers, mediated by the LT toxin, which induced paracellular permeability [120]. In the present study, we showed that only the claudin-1 encoding gene was upregulated following ETEC challenge. Generally, the upregulation of tight-junction-related genes is regarded as beneficial for the host [121,122]. Meanwhile, we could presume that our observation may result from an activation of innate immunity interacting especially with tight junctions following ETEC infection [123,124,125,126]. When fiber-containing products were added, the most remarkable effects were observed with yeast cell walls, which abolished the ETEC induction of *CLDN1* but also significantly decreased transcellular and paracellular permeability and increased TEER values. Up to now, no study conducted on ETEC of human origin has ever reported an attempt to modulate the induced changes in epithelial integrity with dietary-fiber-containing products. Contrarily, in vivo studies in pigs have already shown a beneficial effect on the intestinal barrier disruption of dietary fibers such as chitosan or fructooligosaccharides [127,128,129]. This positive effect may result from a lower innate immunity activation, as reported by decreases in TLR4 and CD14 expression [127,128] and serological cytokines [129]. However, we cannot also exclude a sedimentation effect of the fiber products upon the intestinal cells or a binding with the molecules used as permeability markers. We argue that, at least, the products are unlikely to be detrimental to cellular integrity. Of note, on the contrary, some authors reported detrimental effects of fibers such as cellulose and arabinoxylan [130], indicating that the outcomes are probably fiber-specific.

Evidence from previous in vitro and in vivo studies support an influence of ETEC strains on human gut microbiota [13,14,43,131,132]. As microbiota alterations can even favor enteric infections [133,134], we investigated the impact of ETEC strain H10407 on the gut microbiota structure and activity and how it can be further modulated by supplementation with fiber-containing products. None of the tested products were able to restore microbiota evenness that was, according to human in vivo data, decreased with ETEC infection [14]. We showed that ETEC inoculation was particularly detrimental to mucosal-associated *Clostridium* species, as already reported by Roussel et al. [43]. Supplementation with dietary-fiber-containing products enabled a slight but consistent (in most individuals) maintenance of *Clostridium*. Yeast cell walls also induced ETEC-unrelated changes in microbiota composition, with increases in *Parabacteroides* in both luminal and mucosal compartments. This result deserves more attention since *Parabacteroides* species have already been highlighted as a potential new generation probiotic species in intestinal inflammation-related diseases such as metabolic syndrome [135,136] and colorectal cancer [137]. Up to now, only *Lactobacillaceae* have been regularly highlighted as probiotic species with anti-infectious properties against human ETEC strains [138,139,140]. Here, one donor was particularly colonized by *Lactobacillaceae,* and this bacterial population was found to be enriched on mucin beads by yeast cell walls in the infected condition. Interestingly, this donor was also the one with the lowest proportion of *Escherichia*/*Shigella* on mucin beads. Regarding gut microbial activity, we showed that ETEC inoculation had contradictory effects on fermentation activities, increasing butyric acid production, gas pressure, and CO_2_ level but limited pH acidification. This may result from ETEC mucinase activities, leading to higher substrate availability for fermentation, combined with *E. coli* acid resistance systems, which notably consume H^+^ to produce H_2_O, H_2_, and CO_2_ [141]. Up to now, only two in vitro studies had evaluated the effect of ETEC on human gut microbial activity [43,132]. However, major differences in experimental conditions hampered any comparison. When added, fiber-containing products had a small impact on ETEC-induced changes in microbiota activity. Unsurprisingly, they only seem to favor even more fermentation activities (e.g., fermentation gases). Lastly, since previous studies have elegantly shown in mice that dietary fiber intakes limited pathogen infection by protecting the mucus layer from degradation [36,142,143], we measured the total weight of mucin beads at the end of batch experiments. However, this previous hypothesis was not confirmed here, probably because of the use of simple batch experiments, which did not include goblet cells or allow the continuous supply of fiber sources or a renewal of luminal content.

## 5. Conclusions

Using a large panel of in vitro models, this study demonstrated that fiber-containing products, namely, a lentil extract and yeast cell walls, can exert anti-infectious activities against the human reference strain ETEC H10407. Tested products were found to interfere with the ETEC infection process during virulence gene expression, cell adhesion, cross talk with intestinal host cells, and interactions with gut microbiota. Even if the products were not pure fibers, these results are encouraging for further mechanistic investigations. Next steps should be dedicated to the study of dietary fibers/ETEC interactions in more complex and dynamic multi-compartmental models of the human GI tract, such as the TNO intestinal model (TIM) or the Simulator of the Human Intestinal Microbial Ecosystem (SHIME) before going further in animal models, where we can evaluate their effect on the whole organism (e.g., prevention of diarrhea). These findings reveal important implications regarding how our immediate diet history may modify susceptibility to some enteric diseases but also provide meaningful insights in the use of low-cost dietary-fiber-containing products as a relevant prophylactic strategy in the fight against ETEC infections and traveler’s diarrhea.

## Figures and Tables

**Figure 1 nutrients-14-02146-f001:**
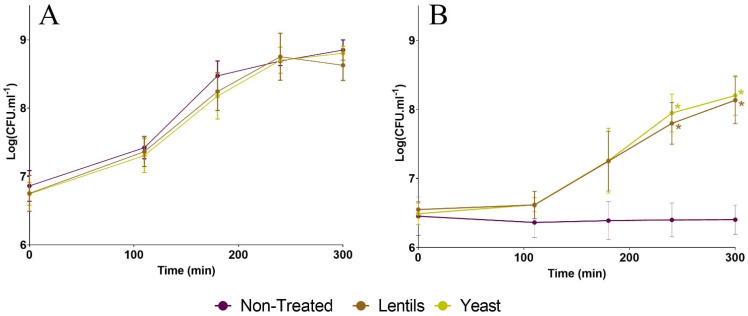
Impact of fiber-containing products on ETEC growth in broth media. Growth kinetics of ETEC strain H10407 (inoculation at 10^7^ CFU·mL^–1^) in LB medium (**A**) or in M9 minimal medium (**B**) supplemented with specific yeast cell walls (yellow line, ‘yeast’), lentils (brown line, ‘lentils’) at 2 g·L^–1^, or not supplemented (purple line, ‘non-treated’). Samples were regularly collected and plated on LB agar. Results are expressed as Log_10_ CFU·mL^–1^ (mean ± SD, *n* = 3). Statistical differences with the control condition are indicated and provided by Dunnett’s multiple comparisons test (*: *p* < 0.05).

**Figure 2 nutrients-14-02146-f002:**
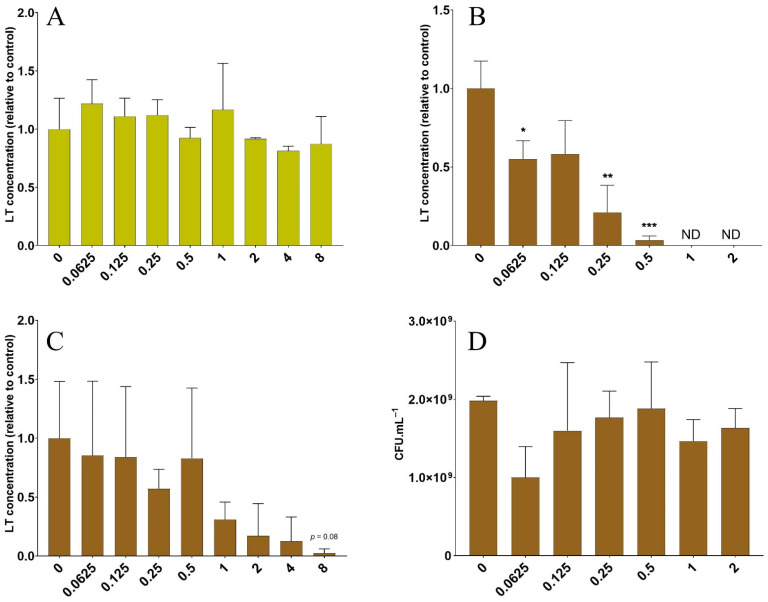
Dose effect of fiber-containing products on LT toxin concentrations in broth media. LT concentrations were measured in CAYE medium after overnight incubation with ETEC strain H10407 (**A**,**B**) or pure LT-Cholera B toxin sub-unit (**C**) and increasing doses of specific yeast cell walls (**A**) or lentil extract (**B**,**C**). Fiber-containing product concentrations are expressed in g·L^−1^ of final fiber content. Results are expressed as mean variations (±SD) compared to the control condition (no product added). The data represent the replicates of three independent experiments. (**D**) The dose effect of the lentil extract on ETEC growth in CAYE medium after overnight incubation. Results are expressed as mean CFU·mL^−1^ (± SD) of three independent replicates. Statistical differences with the non-treated control condition were provided by Tukey’s multiple comparisons tests (*: *p <* 0.05; **: *p <* 0.01; ***: *p <* 0.001). ND = non-detected.

**Figure 3 nutrients-14-02146-f003:**
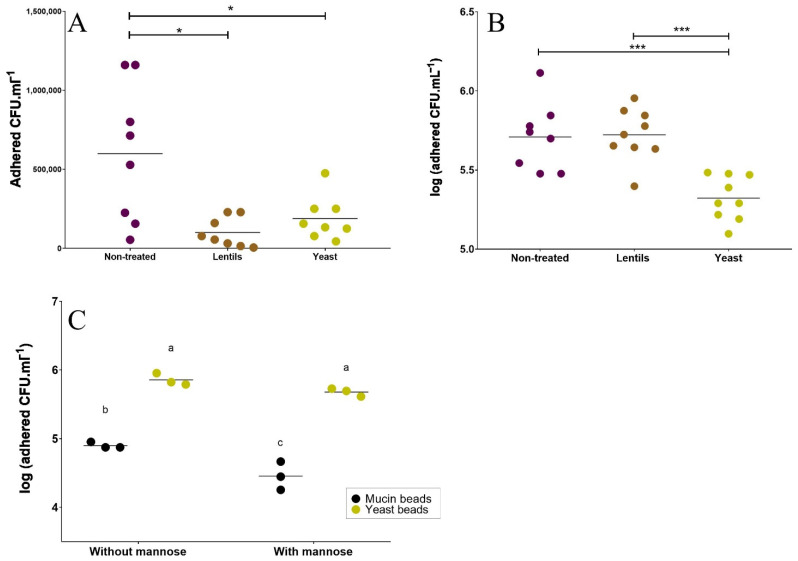
Effects of fiber-containing products on ETEC adhesion on mucin and mucus-secreting intestinal cells. (**A**,**B**) The impact of the lentil extract (brown, ‘lentils’) and specific yeast cell walls (yellow, ‘yeast’) on ETEC adhesion (initial concentration: 10^7^ CFU·mL^−1^) was investigated using two different in vitro assays and compared to the non-treated control condition (purple). (**A**) ETEC adhesion to mucin beads after a 1 h infection period. Results are expressed as adhered cells (CFU·mL^−1^). (**B**) ETEC adhesion to Caco-2/HT29-MTX co-culture model after a 3 h pre-treatment with fiber-containing products followed by an additional 3 h infection period. Results are expressed as adhered cells (Log_10_ CFU·mL^−1^). Figures represent all technical replicates from three independent experiments, and means are indicated by black bars. Indicated *p*-values were provided by Tukey’s multiple comparisons tests (*: *p* < 0.05, ***: *p* < 0.001). (**C**) Adhesion of ETEC strain H10407 (initial concentration: 10^7^ CFU·mL^−1^) on specific yeast–alginate beads (yellow dot) or mucin–alginate beads (black dot), with or without mannose (10 g·L^−1^). Each point represents one of three independent biological replicates and means are indicated by black bars. Results that are not different from each other according to Tukey’s multiple comparisons tests are grouped under the same letter (*p <* 0.05).

**Figure 4 nutrients-14-02146-f004:**
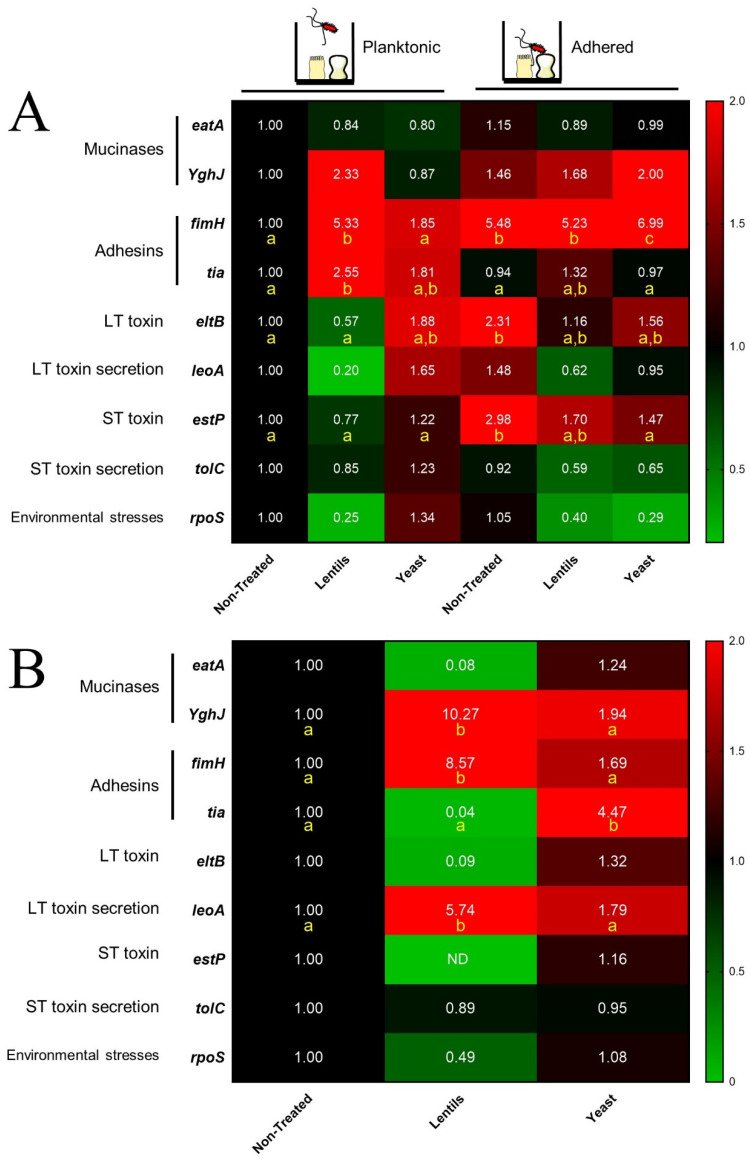
Impact of fiber-containing products on virulence gene expression of planktonic and adherent ETEC cells. ETEC virulence gene expression was analyzed by RT-qPCR in Caco2/HT29-MTX cells infected at MOI 100 (**A**) or in DMEM medium as a control condition devoid of intestinal cells (**B**), with or without lentil extract (‘lentils’) and specific yeast cell walls (‘yeast’) at a final concentration of 2 g·L^−1^. Results are expressed and colored according to fold-change expression compared to the control condition (planktonic bacteria non-treated in (**A**) and non-treated medium in (**B**)). Figure represents at least three independent experiments. If a statistical difference was reached, results that are significantly different from each other according to Tukey’s multiple comparisons tests are grouped under different yellow letters (*p* < 0.05). NT = non-treated, ND = non-detected.

**Figure 5 nutrients-14-02146-f005:**
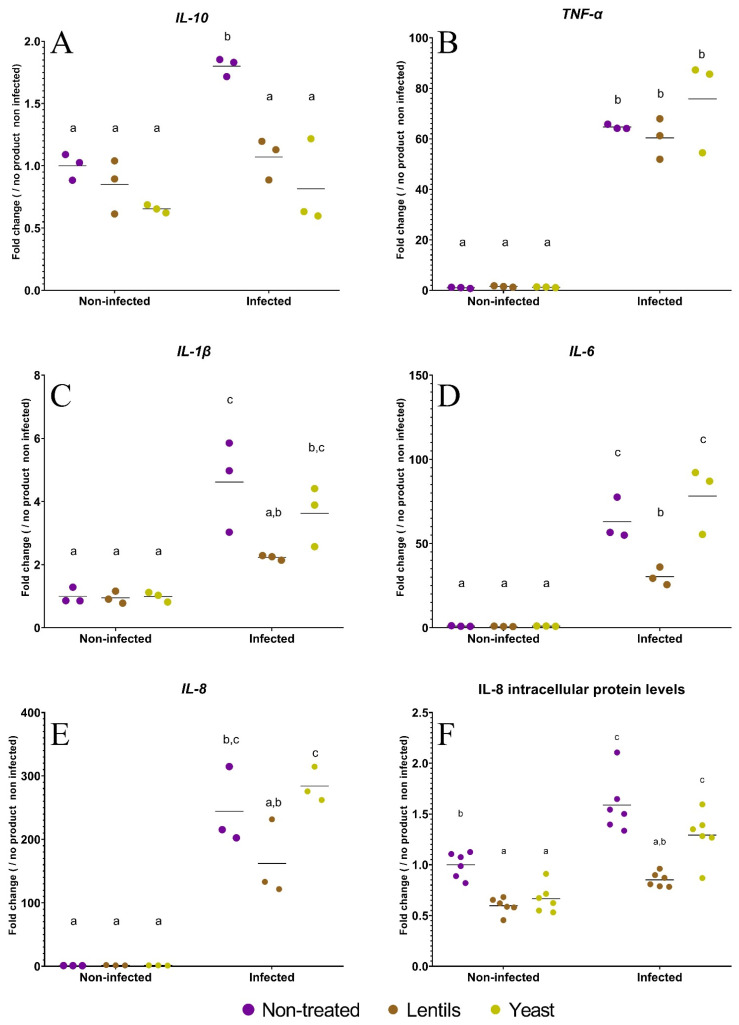
Modulation of host innate immune related genes by fiber-containing products. Caco-2/HT29-MTX cells were infected with ETEC strain H10407 (10^7^ CFU·mL^−1^, MOI 100) after a 3 h pre-treatment with the lentil extract (brown dots, ‘lentils’) or specific yeast cell walls (yellow dots, ‘yeast’). Non-infected and non-treated conditions (purple dots, ‘non-treated’) were used as control experiments. Cytokine (*IL-10*, *TNF-α*, *IL-β*, *IL-6,* and *IL-8*)-related gene expressions were analyzed by RT-qPCR (**A**–**E**), and the interleukin-8 (IL-8) intracellular protein level was measured by an ELISA assay (**F**). Results are expressed as fold changes compared to the non-infected and non-treated control condition. The data represent the replicates of at least three independent experiments with their means. Results that are not different from each other according to Tukey’s multiple comparisons tests are grouped under a same letter (*p <* 0.05).

**Figure 6 nutrients-14-02146-f006:**
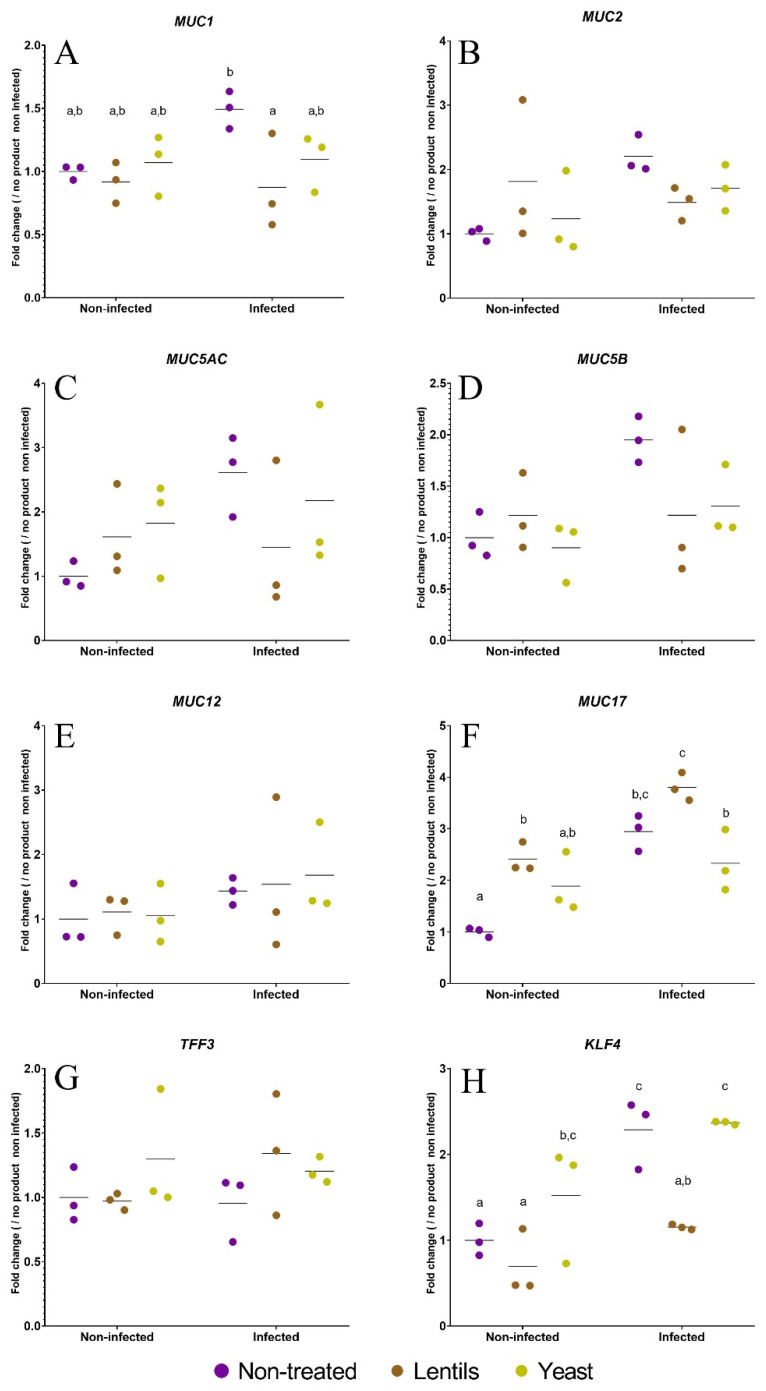
Modulation of mucus-related gene expression by fiber-containing products. Caco-2/HT29-MTX cells were infected with the ETEC strain H10407 (10^7^ CFU·mL^−1^, MOI 100) after a 3 h pre-treatment with the lentil extract (brown dots, ‘lentils’) or specific yeast cell walls (yellow dots, ‘yeast’). Non-infected and non-treated conditions (purple dots, ‘non-treated’) were used as control experiments. The expression of mucus-related genes (*MUC1*, *MUC2*, *MUC5AC*, *MUC5B*, *MUC12, MUC17,* and *TFF3*) (**A**–**G**) and *KLF4* (Kruppel-like factor 4), which is involved in goblet cell differentiation (**H**), were analyzed by RT-qPCR. The results are expressed as fold changes compared to the non-infected and non-treated control condition. The data represent the replicates of three independent experiments with their means. Results that are not different from each other according to Tukey’s multiple comparisons tests are grouped under a same letter (*p <* 0.05).

**Figure 7 nutrients-14-02146-f007:**
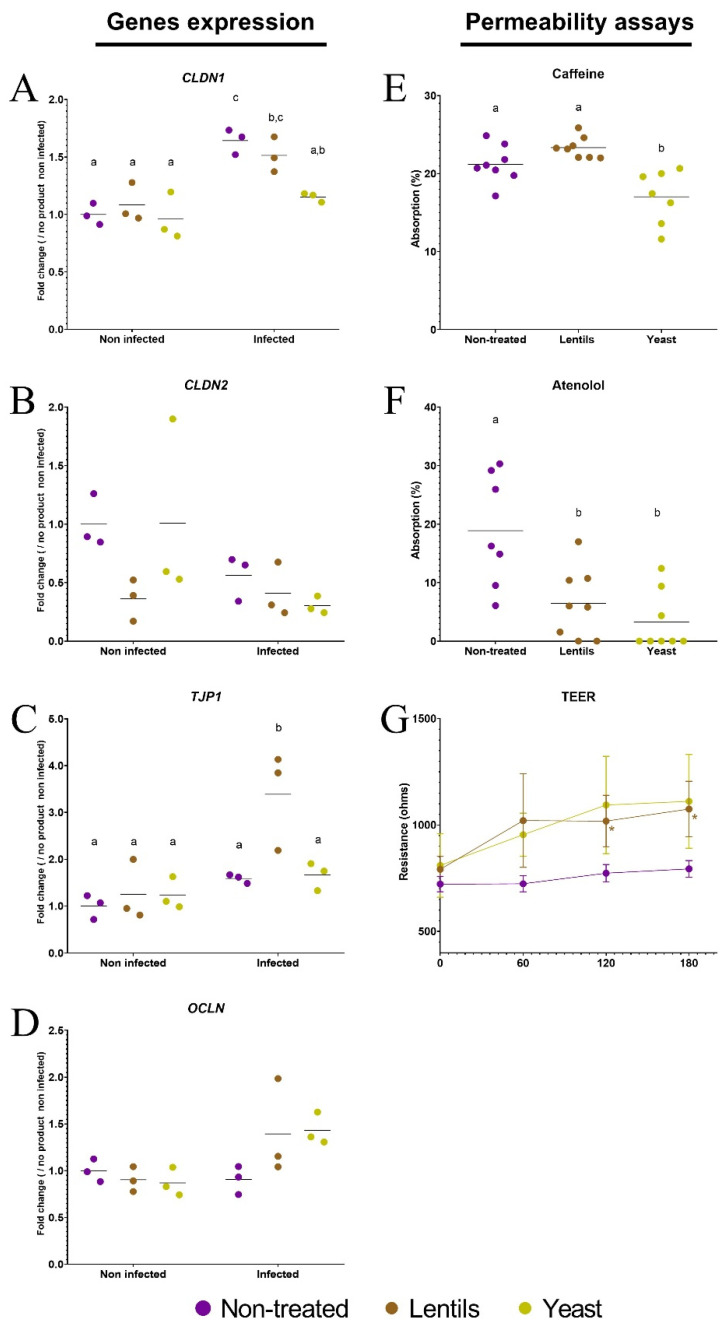
Modulation of intestinal epithelial permeability by fiber-containing products. Caco-2/HT29-MTX cells were infected with the ETEC strain H10407 (10^7^ CFU·mL^−1^, MOI 100) after a 3 h pre-treatment with the lentil extract (brown dots, ‘lentils’) or specific yeast cell walls (yellow dots, ‘yeast’). Non-infected and non-treated conditions (purple dots, ‘non-treated’) were used as control experiments. (**A**–**D**) The expression of tight-junction-related genes (*CLDN1*, *CLDN2*, *TJP1,* and *OCLN*) was analyzed by RT-qPCR. The results are expressed as fold changes compared to the non-infected and non-treated control condition. (**E**,**F**) The absorption of caffeine (1 g·L^−1^) (**E**) and atenolol (50 mg·L^−1^) (**F**) after a 2 h co-incubation of Caco-2/HT29-MTX cells cultured on transwells with or without fiber-containing products. Permeability is given as a percentage of the initial apical concentration. (**G**) Transepithelial resistance (TEER) measured during a 3 h incubation period of Caco-2/HT29-MTX cultured on transwells with or without fiber-containing products. (**A**–**F**) represent individual replicates of three independent experiments with their means, while (**G**) represents the mean resistance (±SD) of three independent experiments. Conditions that are not different from each other according to Tukey’s multiple comparisons tests are grouped under the same letter (*p <* 0.05). In panel (**G**), statistical differences with the non-treated control condition provided by Tukey’s multiple comparisons tests are indicated on the graph (*: *p* < 0.05).

**Figure 8 nutrients-14-02146-f008:**
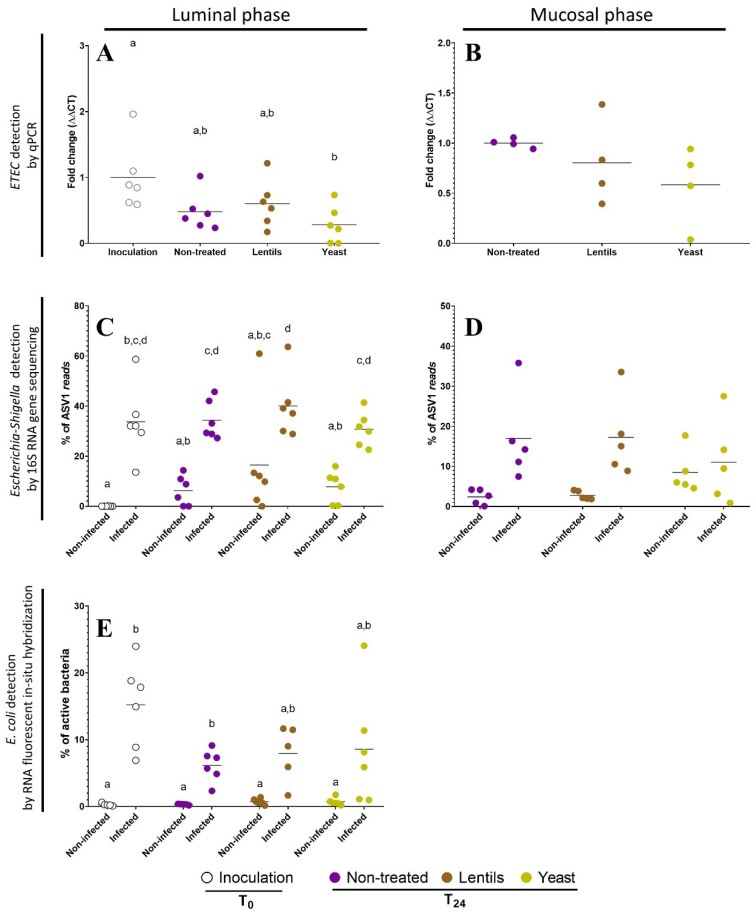
Impact of dietary-fiber-containing products on ETEC survival in in vitro batch colonic conditions. Penicillin bottles containing nutritive medium enriched in dietary-fiber-containing products were inoculated with feces from six healthy donors and then challenged with pre-digested ETEC strain H10407 at 10^8^ CFU·mL^−1^. Control experiments were performed under non-treated and non-infected conditions. White, purple, brown, and yellow dots represent individual biological replicates at the beginning of the experiment after ETEC inoculation (Inoculation, T_0_) or after 24 h of fermentation in the non-treated (Non-treated, T_24_), lentil extract (Lentils, T_24_), or specific yeast cell walls (Yeast, T_24_) conditions, respectively. (**A**,**B**) qPCR detection of the H10407 ETEC strain among the total bacterial population is expressed as fold changes compared to inoculation T_0_ (luminal phase) or non-treated T_24_ (mucosal phase) conditions. (**C**,**D**) Percentages of ASV1 reads detected by 16S RNA gene amplicon sequencing in luminal and mucosal bacteria. ASV1 is the ASV with the highest read abundance in all samples and its reads were assigned to the *Escherichia*/*Shigella* genus and to *Escherichia albertii*/*boydii*/*coli*/*dysenteriae*/*fergusonii*/*flexneri*/*marmotae*/*sonnei* species. (**E**) Proportion of active *E. coli* in the total bacterial populations as detected by RNA fluorescent in situ hybridization. Bars represent the means of data (*n* = 6). Results that are not significantly different from each other according to Tukey’s multiple comparisons tests are grouped under the same letter (*p* < 0.05). ASV: amplicon sequence variant.

**Figure 9 nutrients-14-02146-f009:**
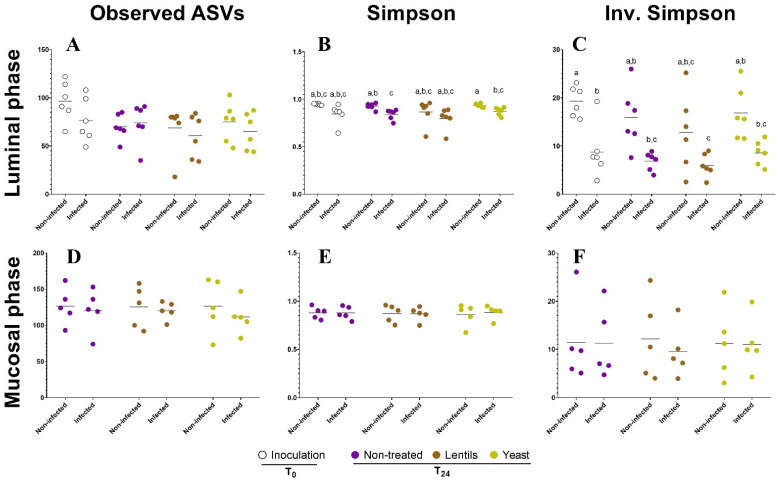
Impact of the dietary-fiber-containing products on the ETEC modulation of microbial community α-diversity. Batch experiments were performed using feces from six healthy donors, challenged or not with ETEC strain H10407 and treated or not with fiber-containing products. The graphs represent the variation in the microbiota species richness (**A**,**D**) and species evenness represented by the Simpson (**B**,**E**) and inverse Simpson indexes (**C**,**F**) at the ASV level. Samples were collected in both the luminal (**A**–**C**) and mucosal compartments (**D**–**F**). White, purple, brown, and yellow dots represent individual biological replicates at the beginning of the experiment after ETEC inoculation (Inoculation, T_0_) or after 24 h of fermentation in the non-treated (Non-treated, T_24_), lentil extract (Lentils, T_24_), or specific yeast cell walls (Yeast, T_24_) conditions, respectively. Black bars represent the means (*n* = 6). Results that are not significantly different from each other according to Tukey’s multiple comparisons tests are grouped under the same letter (*p* < 0.05).

**Figure 10 nutrients-14-02146-f010:**
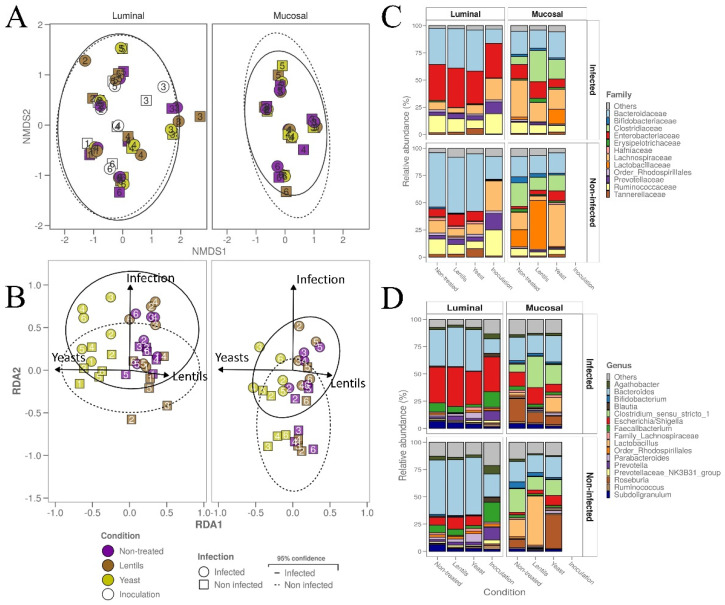
Impact of dietary-fiber-containing products on ETEC modulation of microbial community β-diversity. Batch experiments were performed using feces from six healthy donors, challenged or not with ETEC strain H10407, and treated or not with fiber-containing products. (**A**,**B**) Non-parametric multidimensional scaling (NMDS) (**A**) and distance-based redundancy analysis (db-RDA) (**B**). Two-dimensional plot visualizations report the microbial community β-diversity at the ASV level, as determined by 16S rRNA gene amplicon sequencing. The db-RDA was performed on the ASV table, excluding the inoculation samples (T_0_) and ASV1 (attributed to the *Escherichia*/*Shigella* genus). Infection and fiber products were provided as the sole environmental variables (binary) and are plotted as vectors (arrows). White, purple, brown, and yellow dots represent individual biological replicates at the beginning of the experiment after ETEC inoculation (Inoculation, T_0_) or after 24 h of fermentation in the non-treated (Non-treated, T_24_), lentil extract (Lentils, T_24_), or specific yeast cell walls (Yeast, T_24_) conditions, respectively. The samples are represented by dot shapes and square shapes for the infected and non-infected conditions, respectively. The 95% confidence ellipse area is also indicated in a continuous line for the infected condition and in a dotted line for the non-infected conditions. The donor number is indicated for each sample. (**C**,**D**) Cumulative bar plots of the relative microbial community composition at the family (**C**) and genus (**D**) levels. The area graphs show the relative abundance of the 12 most abundant families and 16 most abundant genera in all six different donors confounded.

**Figure 11 nutrients-14-02146-f011:**
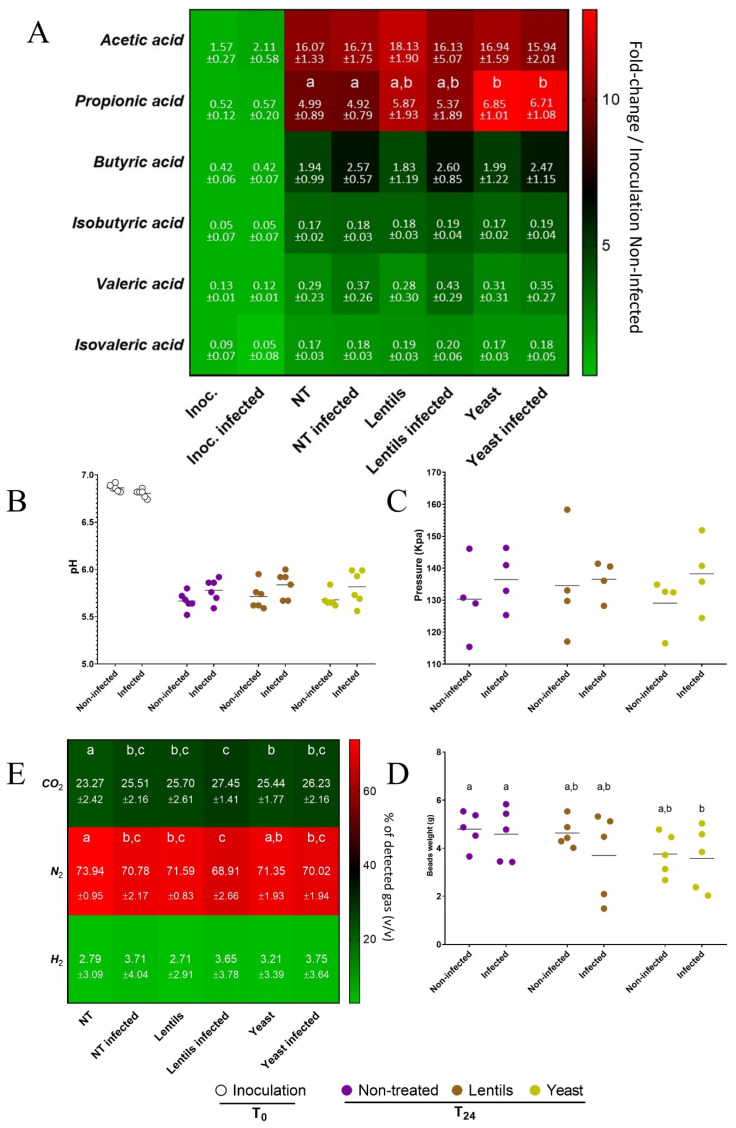
Modulation of gut microbial activity by ETEC infection and dietary-fiber-containing products. The impact of ETEC strain H10407 inoculation (infected versus non-infected) and fiber-containing products (non-treated versus lentils or yeast) on microbiota activity in batch experiments were assayed by the measurement of SCFA production (**A**), pH acidification (**B**), gas pressure (**C**), and gas composition (**D**). Mucin bead weight was also measured at the end of the experiment (**E**). Batch experiments were performed using feces from six healthy donors. White, purple, brown, and yellow dots represent individual biological replicates at the beginning of the experiment after ETEC inoculation (Inoculation, T_0_) or after 24 h of fermentation in the non-treated (Non-treated, T_24_), lentil extract (Lentils, T_24_), or specific yeast cell walls (Yeast, T_24_) conditions, respectively. (**A**) SCFA production in the luminal phase was analyzed by liquid chromatography. Results are expressed in mM ± SD (*n* = 6) and colored according to fold changes compared to the control condition (non-infected, non-treated, T0). (**B**) pH of the fermentation medium was followed-up over-time, and biological replicates are represented as dots with their means (black line). (**C**) Gas pressure was measured at T24h, and biological replicates are represented as dots with their means (black line). (**D**) Gas composition was determined by gas chromatography at T24h. Results are expressed as mean percentages ± SD (*n* = 6) and colored accordingly. (**E**) Mucin beads collected at T24h were weighed, and biological replicates are represented as dot plots with their means (black line). Results that are not significantly different from each other according to Tukey’s multiple comparisons tests are grouped under the same letter (*p* < 0.05). NT: Non-treated.

**Figure 12 nutrients-14-02146-f012:**
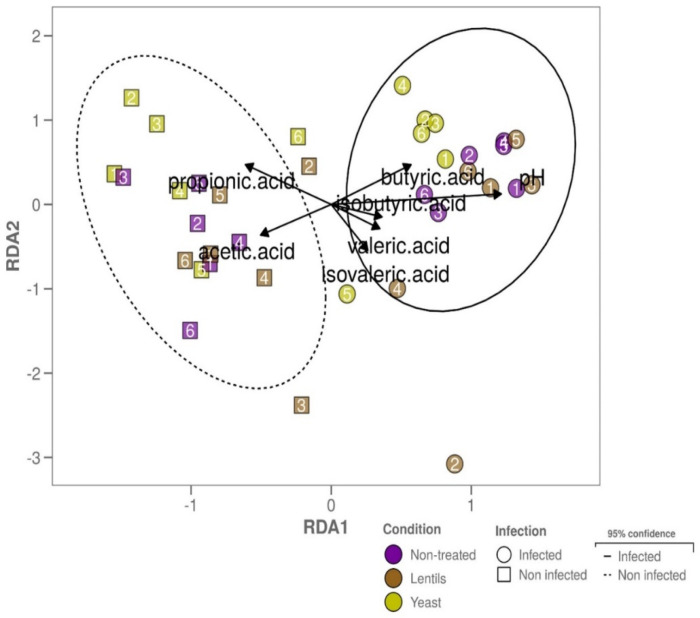
Distance-based redundancy analysis modeling of the microbiota community structure according to pH and SCFA production. The two-dimensional plot reports the β-diversity structure of the whole microbial community taxonomy at the end of the experiment (T_24_) at the ASV level in the luminal phase according to db-RDA and explained by metabolite variables (SCFAs and pH). Individual samples are represented by dot shapes and square shapes for the infected and non-infected conditions, respectively. The 95% confidence ellipse zone is also indicated in a continuous line for the infected condition and in a dotted line for the non-infected conditions. The donor number is indicated for each sample.

**Table 1 nutrients-14-02146-t001:** Detailed composition of the two fiber-containing products that were tested.

Parameters Tested	Results (+/−Incertitude) (g·100 g^−1^)		Analytical Methods
	Lentils	Specific Yeast Cell Wall	
Moisture at 70 °C and low pressure	6.1 (+/−0.5)	7.7 (+/−0.5)	Steaming
Ashes	4.1 (+/−0.2)	3.0 (+/−0.2)	Incineration
Protein “N*6.25”	36.1 (+/−1.0)	13.3 (+/−0.6)	Kjeldahl
Fat “B”	0.90 (+/−0.50)	15.9 (+/−1.3)	Soxhlet
Total carbohydrates	52.7	60.2	Calculation
Total dietary fibers	34.8 (+/−2.0)	57.7 (+/−2.0)	Enzymatic method
Carbohydrates	17.9	2.5	Calculation

**Table 2 nutrients-14-02146-t002:** Primers used in this study.

Gene	Target	Primer Sequence 5′-3′	Amplicon Length (pb)	References
Genes to monitor ETEC survival by qPCR (in fecal batches)				
*eltB*	LT toxin	F-GGCAGGCAAAAGAGAAATGG R-TCCTTCATCCTTTCAATGGCT	117	[44]
*16S*	Reference gene	F-NNNNNNNNNTCCTACGGGNGGCWGCAG R-NNNNNNNNNNTGACTACHVGGGTATCTAAKCC	464	[45]
Genes for RT-qPCR analysis of ETEC virulence genes				
*tufA*	Reference gene	F-GACATGGTTGATGACGAAGA R-GCTCTGGTTCCGGAATGTA	199	[46]
*ihfB*	Reference gene	F-CTGCGAGGCAGCTTCCAGTT R-GCAGCAACAGCAGCCGCTTA	419	[47]
*eltB*	LT toxin	F-GGCAGGCAAAAGAGAAATGG R-TCCTTCATCCTTTCAATGGCT	117	[44]
*leoA*	LT enterotoxin output	F-AAACGGTGCATATCCTCGTC R-AAATGCTGCCACCGAAATAC	168	[43]
*estP*	ST toxin	F-TCTTTCCCCTCTTTTAGTCAG R-ACAGGCAGGATTACAACAAAG	165	[48]
*tolC*	TolC outer membrane protein (ST toxin secretion)	F-AAGCCGAAAAACGCAACCT R-CAGAGTCGGTAAGTGACCATC	101	[49]
*tia*	Tia Adhesin	F-ACAGGCTTTTATGTGACCGGTAA R-GACGGAAGCGCTGGTCAGT	67	[50]
*imH*	Minor component of Type I pili	F-GTGCCAATTCCTCTTACCGTT R-TGGAATAATCGTACCGTTGCG	164	[51]
*yghJ*	Mucinase	F-CCCTGTTAGCCGGTTGTGAT R-GGTATCGGTTCTGGCGTAGG	166	This study
*eatA*	Mucinase	F-AACGGAAGCACCGTCATTCT R-CAGAGTCAGGGAGGCGTTTT	363	This study
*rpoS*	Environmental stresses response	F-GCGCGGTAGAGAAGTTTGAC R-GGCTTATCCAGTTGCTCTGC	229	[52]
Genes for bacterial quantification by RNA fluorescent in situ hybridization in batch fermentation				
*16S*	Eubacteria 16S rRNA	1-GCTGCCTCCCGTAGGAGT 2-CGGCGTCGCTGCGTCAGG 3-MCGCARACTCATCCCCAAA	N/A	[53]
*16S*	*E. coli* 16S rRNA	1-GCAAAGGTATTAACTTTACTCCC *(Cy5 in 5′)* 2-GCAGCAACAGCAGCCGCTTA *(Helper probe)*	N/A	[54]
Genes for RT-qPCR analysis of host response				
**Mucin-related genes**				
*MUC1*	Mucin 1	F-AGACGTCAGCGTGAGTGATG R-CAGCTGCCCGTAGTTCTTTC	172	[55]
*MUC2*	Mucin 2	F-CAGTGTGTCTGTAACGCTGG R-AATCGTTGTGGTCACCCTTG	160	This study
*MUC5AC*	Mucin 5AC	F-GTTTGACGGGAAGCAATACA R-CGATGATGAAGAAGGTTGAGG	278	[56]
*MUC5B*	Mucin 5B	F-GTGACAACCGTGTCGTCCTG R-TGCCGTCAAAGGTGGAATAG	171	[56]
*MUC12*	Mucin 12	F-ACCTTAGCACCAGGGTTGTG R-GGAGGATGCGTCATTCATCT	204	[55]
*MUC17*	Mucin 17	F-TGCAGAACAGGACCTCAGTG R- AGGTCATCTCAGGGTTGGTG	206	[55]
*TFF3*	Trefoil factor 3	F-AGGAGTACGTGGGCCTGTCT R-AAGGTGCATTCTGCTTCCTG	175	[55]
*KLF4*	Kruppel-like-factor 4	F-CTCACCCCACCTTCTTCACC R-AAGGTTTCTCACCTGTGTGG	202	This study
**Tight-junction-related genes**				
*CLDN1*	Claudin 1	F-TGGAAGACGATGAGGTGCA R-AAGGCAGAGAGAAGCAGCA	206	[55]
*CLDN2*	Claudin 2	F-CATTTGTACCTGCAAGGTCTTCT R-GCCTAGGATGTAGCCCACAA	236	This study
*OCLN*	Occludin	F-ACTTCAGGCAGCCTCGTTAC R-CCTGATCCAGTCCTCCTCCA	170	[55]
*TJP1*	Zonula occludens 1	F-GTGCTGGCTTGGTCTGTTTG R-TCTGTACATGCTGGCCAAGG	159	[55]
**Inflammation-related genes**				
*TNF*	Tumor necrosis factor α	F-GCCCATGTTGTAGCAAACCC R-AGGAGGTTGACCTTGGTCTG	242	This study
*IL6*	Interleukin 6	F-CCAGAGCTGTGCAGATGAGTACA R-GGCATTTGTGGTTGGGTCAGG	101	[57]
*IL8*	Interleukin 8	F-TCTGCAGCTCTGTGTGAAGG R-TGAATTCTCAGCCCTCTTCAA	252	This study
*IL10*	Interleukin 10	F-GGCGCTGTCATCGATTTCTTC R-CACTCATGGCTTTGTAGATGCC	108	[57]
*IL1β*	Interleukin 1*β*	F-AGCCATGGCAGAAGTACCTG R-TGGTGGTCGGAGATTCGTAG	171	[58]
**Housekeeping genes**				
*GADPH*	Housekeeping gene	F-GGAGTCCACTGGCGTCTT R-GAGTCCTTCCACGATACCAAA	235	[56]
*HPRT*	Housekeeping gene	F-TTGCTGACCTGCTGGATTAC R-AGTTGAGAGATCATCTCCAC	149	[59]
*PPIA*	Housekeeping gene	F-TGCTGACTGTGGACAACTCG F-TGCAGCGAGAGCACAAAGAT	136	[60]

**Table 3 nutrients-14-02146-t003:** Static in vitro gastro-ileal digestion procedure. A static batch incubation (Erlenmeyer) was used to reproduce the physicochemical parameters of gastro-ileal digestion. Digestive secretions and solutions for pH adjustment were manually added during the 90 min digestion.

Parameters of Static In Vitro Digestion	Gastric Vessel	Duodenum–Ileum Vessels
pH	from 6 (T0) to 2.1	maintained at 6.8
Volume (mL)	50	90
Secretions	(i) 5.36 mg of pepsin (727 U.mg^−1^) (ii) 4.28 mg of lipase (32 U.mg^−1^) (iii) HCl 0.3 M (iv) NaHCO_3_ 0.5 M if necessary	(i) 0.9 g of bile salts (27.9 mM in solution) (ii) 1.8 g of pancreatin 4 USP (iii) Trypsin 2 mg·mL^−1^ (15156 U/mg protein) (iv) NaHCO_3_ 0.5 M if necessary
Time period in batch (min)	30	60
Chyme mixing	100 rpm (magnetic stirrer)	100 rpm (magnetic stirrer)
[Total microbes]	sterile	sterile
Oxygen level (%)	20	20
Temperature (°C)	37	37

## Data Availability

The *16S RNA* gene amplicon sequencing data were deposited and are publicly available in the NCBI Sequence Read Archive database with accession number PRJNA802368.

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
