# Peer review of "Lentils and Yeast Fibers: A New Strategy to Mitigate Enterotoxigenic Escherichia coli (ETEC) Strain H10407 Virulence?"

_nutrients, 2022, doi:10.3390/nu14102146_

Round 1

Reviewer 1 Report

The authors carried out a series of in vitro investigations to reflect the effect of lentil and yeast wall fiber-containing products on Enterotoxigenic Escherichia coli (ETEC) strain H10407 infection, namely through investigation of bacterial growth, adhesion to mucus and intestinal epithelial cells, toxin production and regulation of main virulence genes, impact on intestinal barrier integrity, induction of innate immunity and microbiota modulation. These findings reveal important implications regarding how our immediate diet history may modify susceptibility to some enteric diseases and provide meaningful insights in the use of dietary fibers as a relevant prophylactic strategy in the fight against ETEC infections and traveler’s diarrhea.

The experiments were well done and discussed, the choice of methods was appropriate, and the paper has few grammatical errors detected. The following issues be addressed.

  1. The dosage of the two fiber-containing products used in the investigations were made according to the fiber concentration. Does it mean that the other components in the products had no effect on anti-ETEC infection? In other word, whether the other components should be removed to further verify the function of fiber in mitigating the ETEC virulence? In addition, why there existed differences in capacity to anti-ETEC infection between the two products in the same dosage in your view?
  2. Are the lentils and yeast fibers used in the investigations available on the market? If not, I think the detail methods should be written.
  3. In figure S1, the cell activity was investigated with Caco-2 and HT29-MTX cells alone while the Caco-2/HT29-MTX co-culture (ratio 70:30) was used in experiments of adherence and intestinal permeability on transwell. Does the data in former assay forceful enough to support the latter one? In addition, in figure S1 D, why there existed a significant difference between non-treated group and yeast treated group at the initial time?
  4. In line 141, how to evaluate whether the cells reach the full differentiation stage?
  5. In results 3.5, how to explain that ETEC incubation induced up-regulation of pro-inflammation factors mRNA expression but also anti-inflammation factor IL-10? In addition, the other cytokines achieved significant differences compared with ETEC group should be investigate in protein levels like IL-8.
  6. In figure 7, results of permeability assay in figure 7 E, F&G revealed that both fiber-containing products could strengthen the intestinal barrier function compared with non-treated group seemed to be paradoxical with the results of gene expression in figure 7A, B, C&D which showed no differences between three groups when not infected with ETEC. What’s more, ETEC infection induced relative tight junction and mucin secretion genes up-regulation, which seemed inconsistent with conventional cognition.
  7. Result 3.9 had a mistaken headline the same with 3.8.
  8. Line 924, Doi numbers is missing.

Reviewer 2 Report

In this manuscript, Thomas Sauvaitre investigated the inhibitory effect of two dietary fiber containing-products on E. coli infection. Dietary fibers have well-known beneficial effects for gut health, it is interesting to test the anti-infectious potential of dietary fibers. Following the previous study that select some promising fiber products that influence E. coli growth and LT toxin production, the authors choose two fibers to investigate more deeply about their anti-infectious potential. Unfortunately, they did not move further to reveal the mechanisms e.g., by which the used fibers influence the virulence of E. coli and reduce inflammatory cytokines expression of host cells.  Also, it will increase the significance of this study if these fibers are tested in animal model for anti-infectious potential. The following are some specific comments.

  1. According to your previous study, it seems that the fibers you selected for this study are insoluble, which resists fermentation, it is important to include some soluble fibers which could be fermented by bacteria and compare the difference between insoluble and soluble fiber in anti-infection.
  2. Could you explain how the fiber influence the expression of virulent genes of E. coli? Could E. coli sense the fibers or the other products in the extract that influence their virulent genes expression?
  3. In Figure 5, did you count the number of bacteria when co-cultured with cells lines in presence of fibers, that may cause the difference of cytokine genes expression?

Round 2

Reviewer 2 Report

The authors have addressed my concerns.